# Double-negative B cells and DNASE1L3 colocalise with microbiota in gut-associated lymphoid tissue

Lucia Montorsi[1,12], Michael J. Pitcher [1,12], Yuan Zhao[1], Chiara Dionisi[1], Alicia Demonti [1,2], Thomas J. Tull[3], Pawan Dhami [4], Richard J. Ellis [5], Cynthia Bishop[5], Jeremy D. Sanderson[1,6], Sahil Jain [7], David D'Cruz[1,7], Deena L. Gibbons [1], Thomas H. Winkler [8], Mats Bemark [9,10], Francesca D. Ciccarelli[11] & Jo Spencer [1] ✉

Intestinal homeostasis is maintained by the response of gut-associated lymphoid tissue to bacteria transported across the follicle associated epithelium into the subepithelial dome. The initial response to antigens and how bacteria are handled is incompletely understood. By iterative application of spatial transcriptomics and multiplexed single-cell technologies, we identify that the double negative 2 subset of B cells, previously associated with autoimmune diseases, is present in the subepithelial dome in health. We show that in this location double negative 2 B cells interact with dendritic cells co-expressing the lupus autoantigens DNASE1L3 and C1q and microbicides. We observe that in humans, but not in mice, dendritic cells expressing DNASE1L3 are associated with sampled bacteria but not DNA derived from apoptotic cells. We propose that fundamental features of autoimmune diseases are microbiota-associated, interacting components of normal intestinal immunity.

Gut-associated lymphoid tissue (GALT) is the organised lymphoid tissue on the boundary between the contents of the gut lumen and the host[1]. It is separated from the lumen by the follicle associated epithelium (FAE) that supports the sampling of particulate gut contents by microfold (M) cells and dendritic cells (DC)[1,2]. As a consequence, GALT is chronically activated throughout life[3]. The region of lymphoid tissue below the FAE is the subepithelial dome (SED) where the initial encounter between antigens and the immune system occurs[4–6].

The B cell response to sampled gut antigen, including class switching to IgA in GALT, can occur via T cell (CD40) dependent and independent mechanisms; the latter being supported by receptor ligand systems including A Proliferation Inducing Ligand (APRIL) that binds to its receptors TACI and BCMA[7,8]. The IgA generated has specificity for sampled bacterial antigens, including glycans, that may be shared between bacterial species[9].

Lymphocytes activated in GALT become imprinted; that is, they acquire expression of homing receptors such as α4β7 integrin and chemokine receptors CCR9 and CCR10 that are induced by locally generated retinoic acid. Having acquired the capacity to home, they enter the blood and disseminate widely[10]. The homing receptors allow

[1]School of Immunology and Microbial Sciences, King's College London, London, UK. [2]École Normale Supérieure de Lyon, Claude Bernard Lyon 1 University, Lyon, France. [3]St. John's Institute of Dermatology, King's College London, London, UK. [4]Genomics Research Platform and Single Cell Laboratory at Guy's and St Thomas' NHS Foundation Trust, London, UK. [5]Advanced Cytometry Platform (Flow Core), Research and Development Department at Guy's and St Thomas' NHS Foundation Trust, London, UK. [6]Department of Gastroenterology, Guy's and St Thomas' Foundation Trust, London, UK. [7]Louise Coote Lupus Unit, Guy's and St Thomas' NHS Foundation Trust, London, UK. [8]Division of Genetics, Department of Biology, Friedrich-Alexander-University Erlangen-Nürnberg (FAU), Erlangen, Germany. [9]Department of Translational Medicine – Human Immunology, Lund University, Malmö, Sweden. [10]Department of Clinical Immunology and Transfusion Medicine, Sahlgrenska University Hospital, Gothenburg, Sweden. [11]Cancer Systems Biology Laboratory, The Francis Crick Institute, London, UK. [12]These authors contributed equally: Lucia Montorsi, Michael J. Pitcher. ✉e-mail: jo.spencer@kcl.ac.uk

extravasation into and retention within the intestinal lamina propria, thus seeding large areas of gut with effector cells[11]. The dimeric IgA secreted throughout the gut is transported into the lumen where it interacts with the microbiota maintaining diversity and homeostatic balance[9,12]. In addition to generating IgA plasma cells, the chronically activated state of GALT supports the diversification and propagation of marginal zone B cells that also circulate in blood and become concentrated as a population in the splenic marginal zone[13,14].

Despite the importance of human GALT for maintenance of mucosal and systemic health, very little is known of the tissue microarchitecture and cellular interactions that support its activity. Analysis of B cells subsets in GALT has previously identified naïve B cells (CD27−CD45RB−IgM+IgD+) surrounding germinal centres (GC; CD38 + + CD10 + +). These are in turn are surrounded by populations of marginal zone B cells (MZB; CD27+CD45RB+IgM+IgD+) alongside their precursors (MZP; CD27−CD45RB+IgM+IgD+) and memory cells (CD27+CD45RB+IgD−)[15]. It has been assumed from early work using cellular morphology and enzyme immunohistochemistry that the B cells in the SED, including those that infiltrate the FAE are MZB, though analysis by imaging mass cytometry (IMC) with a small panel of antibodies failed to confirm this[15,16]. The subepithelial region is known to be rich in DC expressing lysozyme (LysoDC)[2,17,18] and APRIL[7,8,19]. However, there is little additional information on the cellular composition of the SED in humans.

Here, we applied multiplexed technologies iteratively to characterise the microanatomy of human GALT. We initially analysed isolated cells from GALT compared to matched blood using mass cytometry and observed that the most enriched B cell subset in GALT is the double negative 2 (DN2; CD27−IgD−CD21loCD11chi) population. This subset, that is associated with interferon driven autoimmunity and infection[20,21], was found to be located in the SED and FAE using imaging mass cytometry. By spatial transcriptomics resolved to single cells we identify that the SED is enriched in expression of the lupus associated autoantigen DNASE1L3[22]. Whereas in mice DNASE1L3 is associated with clearance of apoptotic debris[22,23], we show here that its distribution in humans is inconsistent with this, but rather indicates a role for DNASE1L3 alongside another lupus autoantigen, C1q[24], in handling bacterial debris on the mucosal front line, adjacent to DN2 B cells.

## Results

### Double negative (CD27−IgD−) B cells are enriched in human GALT

Fresh paired cell suspensions of blood and GALT from 5 healthy individuals[15], were analysed using mass cytometry after labelling with metal-tagged antibodies (Supplementary Table 1). CD19+ cells were selected for viSNE dimentionality reduction that was run based on the cell surface expression of B cell lineage markers CD10, CD20, CD21, CD24, CD27, CD38, CD45RB, CD138, IgD, IgM, IgG and IgA (Fig. 1a). Cell similarities were visualised using 'spanning-tree progression analysis of density-normalised events' (SPADE) based on the viSNE coordinates. Nodes were manually grouped into SPADE 'bubbles' according to phenotypes of major B cell subsets (Fig. 1b, c and Supplementary Fig. 1) that were then visualised on the viSNE, and the relative frequencies of cells were represented in pie charts (Fig. 1d, e). In addition to GC cells and plasmablasts (PB)/plasma cells (PC) that were expected to be enriched in GALT compared to blood, we also saw enrichment of double negative B cells, in particular DN2/3 cells, which were differentiated from DN1 by their relatively lower expression of CD21(Fig. 1b, c, f)[21].

We analysed the expression of cell surface markers that were not used in the dimension reduction or clustering algorithms in a heatmap of mean marker expression (Fig. 1g). Notably DN2/3 cells, together with memory B cells, MZB and MZP as already described, were enriched in surface expression of FcRL4 in GALT suggesting proximity to epithelium[15,25,26]. They also expressed lower CD40 and HLA-DR than other B cell subsets suggesting reduced capacity for T cell interaction.

Thus DN2/3 cells were enriched in GALT where their cell surface marker expression suggested epithelial proximity and T cell independence.

### Distribution of B cell subsets in human GALT

To investigate the microanatomic locations of B cell subsets, we analysed Regions Of Interest (ROI) in paraffin sections containing human GALT from 5 donors (ileum 3 donors, 5 ROI; appendix 3 donors, 5 ROI; colon 2 donors, 6 ROI) by IMC, using a panel of metal tagged antibodies (IMC Panel 1 in Supplementary Table 2). All ROI selected were well orientated so that each included GC and FAE. Following segmentation, quality control, and identification of major cell subtypes in GALT by machine learning (Supplementary Fig. 2a−c), B cells were identified by their expression of CD20, including those that infiltrate between epithelial cells in the FAE (Fig. 2a and Supplementary Fig. 2f). B cell boundaries were visualised on the images (Fig. 2b). Dimensionality reduction and clustering were performed on the B cells using the B lineage associated markers in the panel and clusters were merged and classified according to marker expression (Fig. 2c, d and Supplementary Fig. 2d, e). The distance of all B cell subsets to the epithelium was measured, and DN2 B cells were found to consistently have shorter distances across all ROIs (Fig. 2e). The B cell subsets were then visualised on the images (Fig. 2f). As expected, GC B cells were surrounded by naïve B cells, MZP, MZB and memory B cells in concentric circles[15]. Two clusters of DN cells were observed; one in the epithelium that carried the E-cadherin marker from adjacent epithelial cells termed intraepithelial DN (IEDN) cells and DN2 cells in the SED that were high in CD11c (Fig. 2d−g). Members of all B cell clusters were also observed in the GC (Fig. 2f).

Because of the close proximity of DN cells to the CD11c expressing dendritic cells (DC) in the SED, we confirmed that B cells in the SED expressed CD11c using an RNAscope probe against ITGAX (gene encoding CD11c), visualised by confocal microscopy (Fig. 2h). By this method we were also able to confirm that a subset of IEDN cells were also DN2. The relative distributions of B cell subsets described above including the dominance of DN2 cells in the SED and FAE was observed not only in GALT from colon, but also in ileum and appendix (Fig. 2f and Supplementary Fig. 2g, h). To further validate the outcomes, further samples from 4 donors (ileum 1 donor, 1 ROI; appendix 3 donors, 3 ROI) were examined manually and the presence of CD11c+ DN B cells adjacent to and within the epithelium was confirmed (Supplementary Fig. 2i)

Thus DN cells, in particular DN2, are found on the mucosal front line in the SED and FAE of GALT. DN2 cells are known to be enriched in blood in lupus and in blood and tissues in Sjögren's disease[20,27].

### Interrogation of the subepithelial dome by spatial transcriptomics

Having identified enrichment of DN2 cells in GALT and their localisation in the SED, we were interested to uncover features of that microenvironment that may influence their function. To that end we used 10X Genomics Visium spatial transcriptomics to identify features of the SED in GALT from human colon (n = 5 GALT sites from 2 donors). In these, we first confirmed that the distribution of genes MS4A1 (CD20), IGHD and IGHM, known to be associated with lymphoid tissue, matched their expected patterns of expression (Fig. 3a).

We used three analysis methods to characterise genes selectively expressed in the SED to better understand the microenvironment in which DN2 cells reside. First, we manually selected the spots that corresponded to the lymphoid tissue by reference to the haematoxylin and eosin (H&E) staining, and within this, separated those belonging to the SED and the follicle (Fig. 3a, b and Supplementary Fig. 3a, b). Due to the resolution of this technique, spots overlaying the FAE were included in the SED group for the analysis. We then used the Seurat package to find markers that differed between the SED and the rest of the

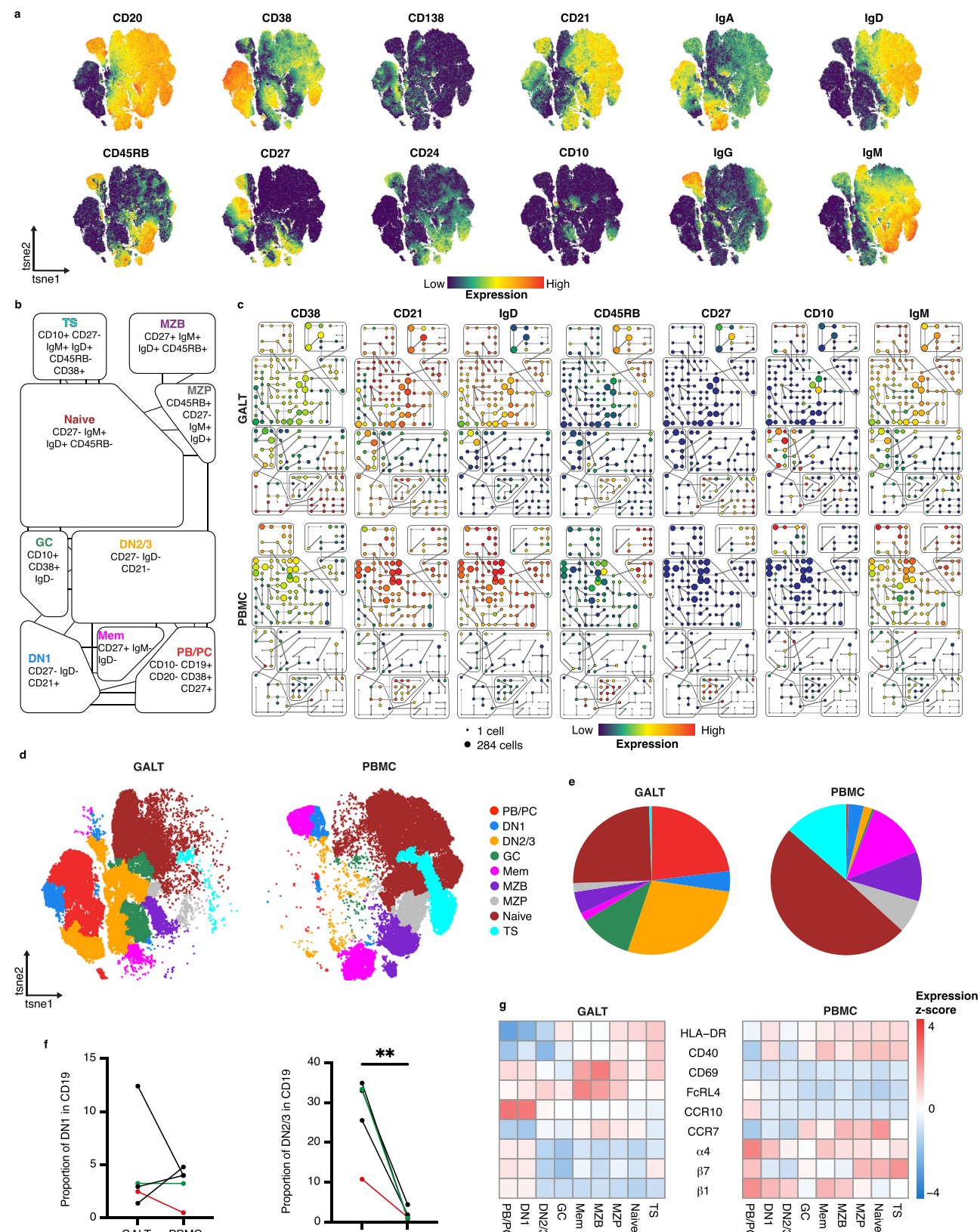

lymphoid tissue and expressed it as a heatmap (Fig. 3c)[28]. Among the most differentially expressed genes were *CCL20* and *CCL23* that are associated with the FAE in mouse models, and *ITGAX* that is expressed by DC in the SED, which confirmed that our approach was successful[29]. Providing further validation, other lymphoid markers included *BCL6* and *TCL1A* identifying the GC and naïve B cells were instead

preferentially expressed in the follicle area[30,31]. Interestingly, the most differentially expressed gene between the SED and the rest of the follicle was *DNASE1L3*, with the complement *C1Q*, *C3* and *C1R* also enriched in the SED (Fig. 3c, d and Supplementary Table 3).

We then undertook an undirected approach to analysis. Spots across the whole section were clustered based on their transcriptomic

**Fig. 1 | Comparison of B cells in GALT and PBMC using mass cytometry. a** viSNE plot of B cells (gated CD19+) from paired gut associated lymphoid tissue (GALT) and peripheral blood mononuclear cells (PBMC) from 5 donors combined, showing expression of markers used for dimensionality reduction. **b** Overview of B cell subset classifications and phenotypes used in SPADE analysis (TS = transitional, MZP = marginal zone precursor, MZB = marginal zone B cells, GC = germinal centre, Mem = memory, PB/PC = plasmablast/plasma cells, DN = double negative). **c** Representative SPADE plots of matched GALT (top row) and PBMC (bottom row) from the same donor. Node size indicates number of cells and colour indicates median expression level for each marker. **d** viSNE plot showing distribution of B cell subsets for all donors, split by GALT (left) and PBMC (right). **e** Pie charts showing relative proportions of each B cell subtype for all donors combined for GALT (left) and PBMC (right). Colouring matches that shown in (**d**). **f** Matched relative proportions of DN1 (left) and DN2/3 (right) for each sample. Statistics between groups were assessed by paired two-tailed t test, $P = 0.5864$ and $0.0043$ respectively, $n = 5$ donors. Red identifies the sample shown in (**b**), and green identifies another sample shown in Supplementary Fig. 1. **g** Heatmap showing the mean expression (scaled by row) of key markers for each subset in pooled samples split by GALT (left) and PBMC (right). Source data are provided as a Source Data file.

profiles and location, and for each image the cluster that coincided with lymphoid tissue by reference to the H&E staining was selected (Fig. 3e, f; and Supplementary Fig. 3d, e). The lymphoid tissue clusters from different images were then subclustered to generate 5 regional clusters within the lymphoid tissue (Fig. 3g). These were classified according to expression of lineage related genes (Fig. 3g, h; and Supplementary Fig. 3g and Supplementary Table 3). We again observed enrichment of *DNASE1L3* expression in the subcluster that equated to the SED/epithelium, and to a lesser extent within a subcluster corresponding to the T cell zone (Fig. 3h). Complement components *C1Q, C3* and *C1R* had a similar distribution to *DNASE1L3*, with enrichment within SED and the T cell zone (Fig. 3h).

DNASE1L3 is a secreted DNASE that is also a recently described lupus autoantigen[22,23]. In addition, loss of function mutations in *DNASE1L3* result in the generation of anti-DNA antibodies. Like *DNASE1L3*, loss of function mutations in *C1Q* can predispose to the production of anti-DNA antibodies and C1q can also be a target of autoantibodies in lupus[32–34]. In contrast, C3 is associated differently with lupus: plasma C3 levels inversely correlate with disease state in SLE and antibodies are occasionally seen against one of its breakdown products[35]. These features were of particular interest to us having observed the lupus associated DN2 B cell subset in this microenvironment. We therefore asked whether expression of selected SLE associated genes *C1QB, C1QA, C1R, C1S, C3, DNASE1L3, ACP2, AGER, CTBB, DNASE1, FAS* and *TMEM173*[36–40] generally spatially overlapped across the tissue irrespective of zonation (Fig. 3i and Supplementary Fig. 3h, i). Expression of TLR7 and TLR9 was too low to be included. Whilst there was strong colocalisation of *DNASE1L3* and *C1Q*, and to some extent *C3* and *C1R*, there was no colocalisation between *DNASE1L3* and the other lupus related genes, including genes associated with apoptosis (Fig. 3i).

Spatial transcriptomics therefore identified enrichment of genes associated in different ways with lupus nephritis in the SED of GALT. Loss of function mutations in *DNASE1L3*[32] or *C1Q*[34] result in the production of antibodies to DNA. They are also lupus associated autoantigens[22,24]. In contrast C3 and C1r are consumed in active lupus, resulting in lower serum levels[35].

**DNASE1L3 expression by DC in the subepithelial region of GALT**
To validate the observations in Fig. 2 and to characterise the cells producing DNASE1L3 in SED, we investigated the expression of *DNASE1L3* transcript in single cells using RNAscope combined with IMC in 5 donors (ileum 3 donors, 4 ROI; appendix 3 donors, 5 ROI; colon 2 donors, 7 ROI) using IMC Panel 2 (Supplementary Table 2). Some antigenic targets, including CD11c, were damaged by the RNAscope tissue processing, thus the distribution of CD11c expression was visualised using the corresponding RNAscope probe (*ITGAX*). Consistent with the spatial transcriptomic analysis, we observed abundant *DNASE1L3* expression in the SED (Fig. 4a–c; Supplementary Fig. 4a–c). Furthermore, the technique allowed us to demonstrate that *DNASE1L3* was expressed by *ITGAX*+CD68+ DC cells in the SED across all donors and ROI (Fig. 4a–c). Expression by DC with proximity to epithelium was confirmed computationally across the 14 ROI with an FAE longer than 100 μm (Fig. 4d).

Surprisingly, *DNASE1L3* expression was absent in the GC from GALT (Fig. 4a–c)[23,41]. This was confirmed computationally across all ROI with GC ($n = 10$) (Fig. 4e and Supplementary Fig. 4a–c). Lack of expression of *DNASE1L3* in GC was also confirmed in 2 ROIs from each of 2 mesenteric lymph node (2 donors) and 1 ROI from each of 2 tonsils (2 donors) analysed (Supplementary Fig. 5a, b). Since the GC is a known site of high apoptotic activity, these observations are in contrast with the accepted function of DNASE1L3, which is to degrade DNA from apoptotic cells[42]. Occasionally, *DNASE1L3* expression was observed in the T cell zone in sparse *ITGAX*+CD68+ cells consistent with other work[43], though at a lower level of expression than in the SED (Fig. 4f).

Due to the location of *DNASE1L3*+ cells in the SED, which is known to be a site of antigen sampling, and their absence from GC we hypothesised that *DNASE1L3* might be associated with anti-microbial activity[17]. Consistent with this hypothesis, *DNASE1L3* expressing cells in this region also expressed anti-microbial enzyme lysozyme and the superoxide generating enzyme NOX2 that is also associated with microbial killing (Fig. 4g, h). *DNASE1L3*+ cells in the SED had a greater tendency to express lysozyme or NOX2 than the less frequent *DNASE1L3*+ cells located elsewhere in the tissues (Fig. 4g–i).

Having observed DN2 B cells in the SED and DC expressing *DNASE1L3* in the same microenviroment, we asked whether these cell types might interact. By RNAscope coupled with confocal microscopy we observed interactions between *ITGAX*+CD11c+CD20+ B cells, likely DN2, and CD11c+*ITGAX*+ dendritic cells in the SED (3 appendix donors, 6 ROIs) (Fig. 4j).

Thus DC that express *DNASE1L3* interact with DN2 B cells in the SED, while expression of the enzyme could not be detected in the GC.

**DNASE1L3 expression is not associated with apoptotic debris in humans, though it is in mice**
The enrichment of DNASE1L3 in the SED and its absence from the GC was surprising, because the most commonly described role for DNASE1L3 is the removal of apoptotic debris, including that generated by B cell selection in the GC response[23,41,42,44]. To ask whether the SED is an area enriched in apoptotic debris in humans, we incorporated the TUNEL assay into IMC Panel 1 (Supplementary Table 2). To overcome the incompatibility of the RNAscope protocol and the TUNEL protocol we applied IMC Panel 1 to sections serial to those that identified *DNASE1L3* distribution by RNAscope with IMC Panel 2 (Supplementary Table 2). Data from ROI containing GALT from 5 donors (ileum 3 donors, 5 ROI; appendix 3 donors, 5 ROI; colon 2 donors 6 ROI) were acquired. We observed large cells in the GC with abundant cytoplasmic accumulation of TUNEL+ material that equated to GC tingible body macrophages, which are known to uptake apoptotic debris (Fig. 5a–c and Supplementary Fig. 6a, b). TUNEL+ macrophages were CD68+ and either CD11c+ (predominantly in the light zone) or CD163+ (predominantly in the dark zone). The epithelium of the gut is known to be constantly renewed, but this happens by extrusion of epithelial cells into the gut lumen[45]. Consistent with this, TUNEL signal and efferocytosis were not features of the SED of GALT, where DNASE1L3 is expressed (Fig. 5a–c; Supplementary Fig. 6a, b).

Much of the evidence for the role of DNASE1L3 in digestion of apoptotic debris in vivo includes murine models[22,23,44]. We therefore

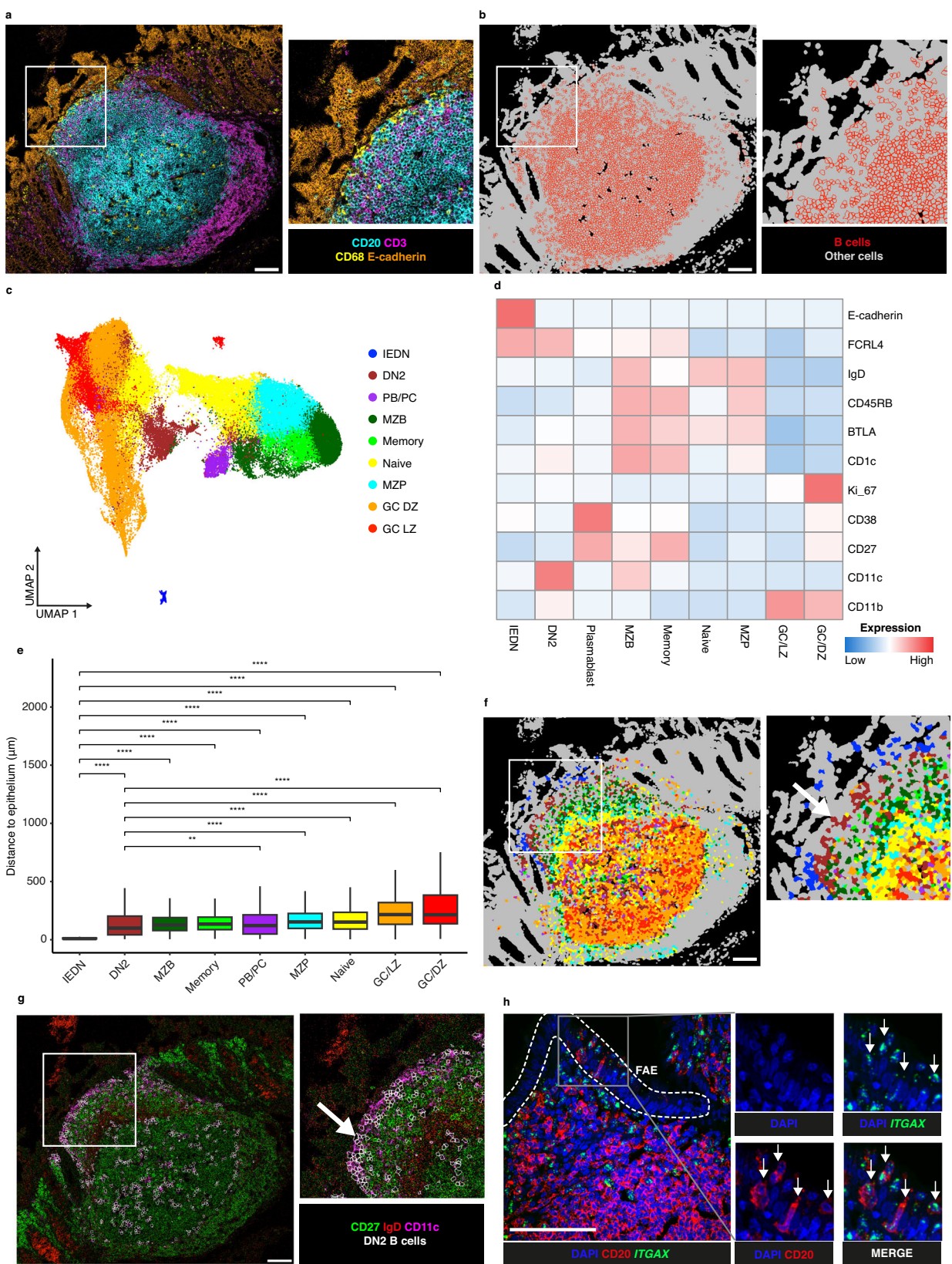

analysed the distribution of *DNASE1L3* in mouse GALT represented by Peyer's patches (*n* = 5 C57BL/6 mice and *n* = 2 CBA mice, at least one Peyer's patch per mouse) using RNAscope and confocal microscopy. In Peyer's patch GCs we observed a marked difference in the expression of *DNASE1L3* in mouse and human GALT. Whilst *DNASE1L3* transcripts were not detected in human GALT GC (Fig. 4), it was expressed in both

the SED and in the GC in mice where it colocalised with lysozyme (Fig. 5d). This suggests that *DNASE1L3* expressing cells in the SED in mice were predominantly LysoDC[17]. The GC cells that expressed *DNASE1L3* and lysozyme in mice also contained pycnotic nuclear fragments (Fig. 5d). Through the use of the TUNEL assay we show that the lysozyme+ cells in the GC that contained pycnotic nuclear

**Fig. 2 | Analysis of B cells in GALT using imaging mass cytometry. a** A representative example of the 16 ROI analysed for this figure. Example is gut-associated lymphoid tissue from human colon showing key lineage marker expression, with inset of epithelium and subepithelial dome region. Cyan = CD20, magenta = CD3, yellow = CD68, orange = E-cadherin. **b** Segmentation of cells (grey) from same ROI as shown in (**a**), with cells computationally classified as B cells highlighted (red). **c** UMAP of B cells with assigned subtypes (DN = double negative, IEDN = intraepithelial DN, PB/PC = plasmablast/plasma cells, MZB = marginal zone B cells, MZP = marginal zone precursor, GC LZ = germinal centre light zone, GC DZ = germinal centre dark zone). **d** Heatmap of mean expression of markers used for dimensionality reduction for B cell subtypes. **e** Distance of B cell subtypes from nearest epithelial cell (in μm). Centre line = median; box limits = upper and lower quartiles;

whiskers = 1.5x interquartile range. Statistics between groups were assessed by two-tailed t test: **$P < 0.01$, ****$P < 0.0001$, $n = 16$ ROIs. Exact $p$ values can be found in the Source Data file. **f** B cell subtypes overlaid on ROI from (**a**, **b**). **g** Imaging mass cytometry derived image of ROI from (**a**), showing DN-related markers CD27 (green), IgD (red) and CD11c (magenta). Computationally derived borders of DN2 cells (white) are overlaid and show that they are found in a region that is CD27-IgD-CD11c+ near the epithelium. **h** Confocal microscopy image of RNAscope experiment showing the distribution of DAPI (blue), CD20 protein (red, pseudocolor) and *ITGAX* transcript (green) in a representative appendix follicle. CD20+*ITGAX*+ cells in the Follicle Associated Epithelium (FAE) are highlighted with arrows. Scalebars = 100 μm. Source data are provided as a Source Data file.

fragments also displayed strong TUNEL signal in the cytoplasm (Fig. 5e), confirming that these were tingible body macrophages performing efferocytosis ($n = 5$ C57BL/6 mice and $n = 2$ CBA mice, at least one Peyer's patch per mouse). Although the majority of efferocytic cells were concentrated in the GC of mice, LysoDC with cytoplasmic TUNEL staining were occasionally observed in the SED (Fig. 5e) which contrasted with what we observed in humans (Fig. 5a–c).

Thus we identified major differences in the distributions of DNASE1L3 in humans and mice. In humans the distribution of DNASE1L3 is inconsistent with a major function in digestion of apoptotic debris in GC.

### Expression of DNASE1L3 by lamina propria subepithelial macrophages

Because of the observed expression of *DNASE1L3* in GALT proximal to the epithelial boundary, and its known inducibility[46], we asked whether lamina propria macrophages might also express *DNASE1L3* in 6 donors (ileum 3 donors, 3 ROI; appendix 4 donors, 5 ROI; colon 2 donors, 8 ROI). We observed lamina propria macrophages that were predominantly *ITGAX* -CD163+CD68+ could express DNASE1L3 (Fig. 6a, b).

In samples from 4 donors (appendix 2 donors, 2 ROI; colon 2 donors, 2 ROI) in which the crypts were well orientated in longitudinal section, we were able to observe that macrophages with *ITGAX* -CD163+CD68+ phenotype expressed *DNASE1L3* when in subepithelial locations closer to the intestinal lumen but only rarely when adjacent to the bases of the crypts (Fig. 6c, d). The difference in phenotype between cells expressing *DNASE1L3* in lamina propria and SED suggests that *DNASE1L3* expression is associated with subepithelial location rather than myeloid cell phenotype or lineage. This is consistent with its previously observed inducibility[46].

### Relationship between expression of DNASE1L3 and complement components

Expression of *C1Q*, like *DNASE1L3*, was found to be enriched in the SED by spatial transcriptomics (Fig. 3c, d) and they appeared spatially associated (Fig. 3h, i). We therefore asked whether *DNASE1L3* and *C1Q* were co-expressed by the same cells in GALT. Using RNAscope coupled with confocal microscopy, we observed co-expression of these two transcripts in the SED in 4 donors (appendix 2 donors, 3 ROI including GC; colon 2 donors, 3 ROI including GC) (Fig. 7a and Supplementary Fig. 6c). Some GC cells were weakly positive for *C1Q* but all were negative for *DNASE1L3* (Fig. 7a).

Complement components *C3* and *C1R* were also identified by spatial transcriptomics to be expressed in the lymphoid tissues in the SED and the periphery of the follicle. RNAscope analysis revealed that *C3* and *C1R* were co-expressed but in different cells to those expressing *C1Q* and consequently *DNASE1L3* (Fig. 7b, Supplementary Fig. 6d) in 4 donors (appendix 2 donors, 5 ROI; colon 2 donors, 4 ROI). *C3* and *C1R* were also found to be co-expressed with vimentin that is expressed by stromal cells (Fig. 7c) in 4 donors (appendix 2 donors, 2 ROI; colon 2 donors, 2 ROI).

### DNASE1L3 expression relative to the distribution of bacteria in GALT

The FAE of GALT and associated DC are known to sample particulate antigens, including bacteria, from the gut lumen, and DNASE1L3 enrichment could be related to sampling or digestion of microbial DNA[2,6,17]. Due to the high immunogenicity of bacterial DNA and shared epitopes with mammalian DNA, rapid degradation of bacterial DNA would be important in this location to avoid potential pathogenic anti-DNA antibody formation, possibly involving the adjacent DN2 B cells.

To assess the presence of bacteria in the SED, we initially used an RNAscope probe against a region of the 16S transcript which is conserved across all Eubacteria in 5 donors (ileum 1 donor, 2 ROI; appendix 2 donors, 2 ROI; colon 2 donors, 2 ROI). Bacterial RNA was detected in the FAE and the SED most proximal to the epithelium (Fig. 8a, b), in the proximity to *DNASE1L3*+ cells, while it was not detected in the rest of the follicle below.

To investigate this further we incorporated the same RNAscope probe into IMC panel 3 that also included major lineage markers (Supplementary Table 2) in samples from 3 donors (ileum 1 donor, 4 ROI; colon 2 donors, 4 ROI). Bacterial RNA in proximity with the FAE appeared to mostly be localised in the extracellular space. However, they were sometimes colocalised with E-cadherin, which is suggestive of bacterial transport into the SED (Fig. 8c, d), an activity typically ascribed to M cells[6]. Bacterial RNA was also often colocalized with CD68 and *DNASE1L3*, while they were not observed to colocalise with other cell types (Fig. 8f, g). This is consistent with uptake of bacteria by *DNASE1L3*+ DC in the SED[47,48], though bacteria can also remain extracellular, most likely temporarily.

To assess whether these bacteria were likely to be intact, we used high resolution confocal microscopy to visualise bacterial DNA stained by DAPI in samples from 3 donors (ileum 1 donors, 2 ROI; colon 2 donors, 6 ROI). Most bacteria detected by RNAscope appeared to also contain DNA suggesting that these are intact cells (Fig. 8h, Supplementary Movie 1, 2). To then consider if these could be members of the normal gut microbiota we used probes against *Escherichia coli* (*E. coli*) and *Bacteroides fragilis* (*B. fragilis*) species and could see examples of these in GALT from 6 donors (ileum 2 donors, 2 ROI; appendix 2 donors, 2 ROI; colon 3 donors, 3 ROI) (Fig. 8i, j).

Thus we observe bacteria in GALT in the FAE, the subepithelial extracellular space and inside *DNASE1L3* expressing DC that are likely to be microbiota derived from the gut lumen.

## Discussion
Here we identify that DN2 B cells are abundant at the boundary between the microbiota and the host in the SED and FAE regions of GALT. These sites are enriched in APRIL that supports the T cell independent B-cell response and class switching towards IgA[8,18]. Consistent with T cell independence, GALT DN cells have relatively low expression of CD40 and class II MHC that are associated with cognate T cell help[7,8]. DN2 are enriched in the blood in lupus nephritis where they can be precursors of autoreactive plasma cells driven by IFNγ[20,49]. This pathway could be active in GALT that is known to be rich in IFNγ

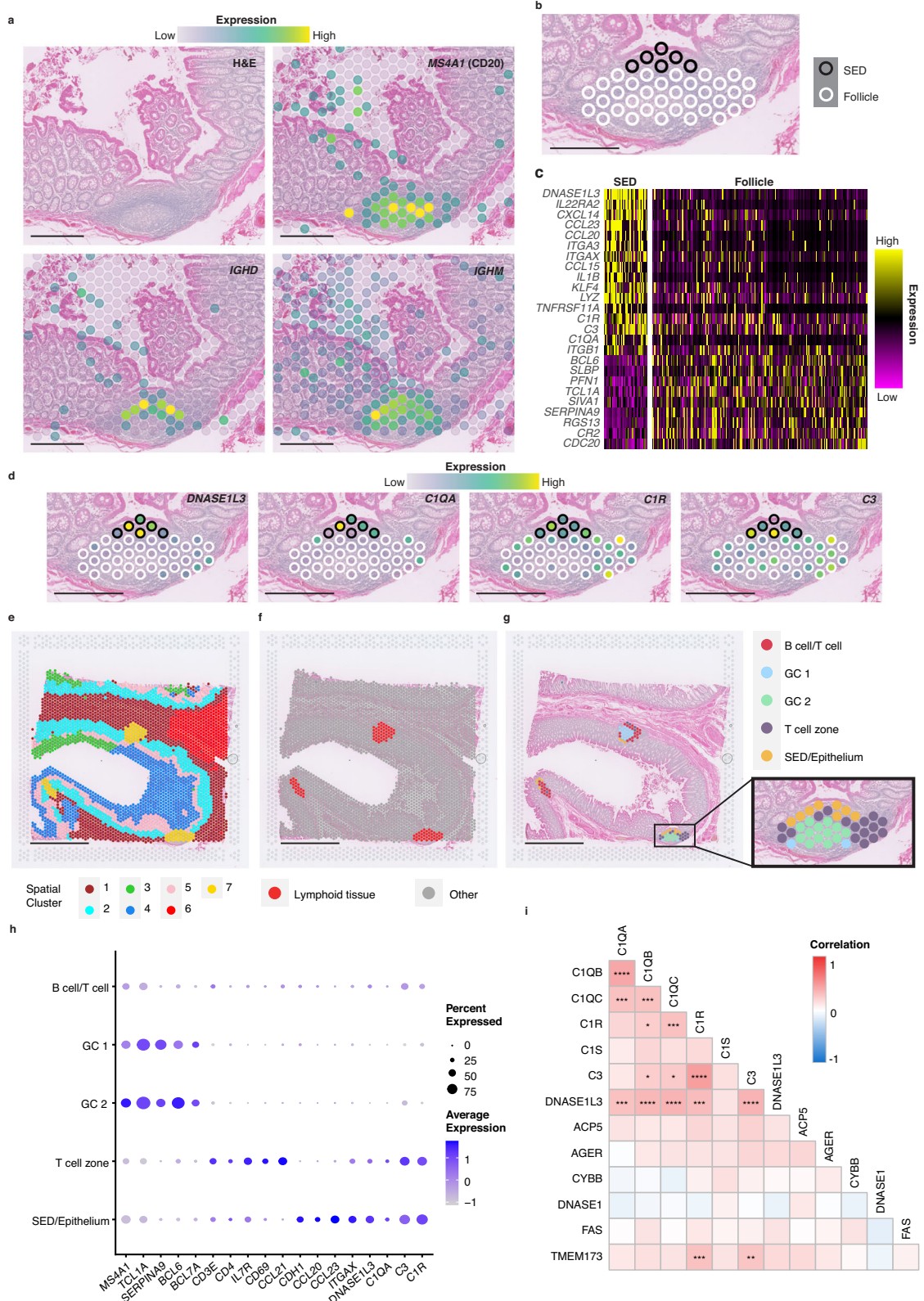

producing cells[50] and to generate IgA via T cell independent pathways[8,51].

We show that DN2 B cells are intermixed with DC that co-express *DNASE1L3* and *C1Q*. Because of their enrichment in the SED in a region between sampled microbiota and the host we propose that DNASE1L3 and C1q are involved in management of bacteria sampled from the gut lumen. In particular, potential for high concentration of secreted

DNASE1L3 in the SED is consistent with it having a role in digestion of bacterial DNA as observed previously in other contexts[52]. Bacterial DNA could be potentially pathogenic and could drive aberrant activity of DN2 B cells that are situated together with the *DNASE1L3*-expressing DC[20].

Whilst a potential role of DNASE1L3 in digesting bacterial DNA in the gut is relatively straight forward to envisage, the potential role of

**Fig. 3 | Spatial transcriptomics analysis of GALT. a** Hematoxylin and Eosin (H&E) staining of GALT in colon, along with the expression of B cell markers in spatial transcriptomics spots showing follicle location. **b** Manual classification of spots as subepithelial dome (SED) (black) or rest of follicle (white). **c** Heatmap of genes differentially expressed between SED and Follicle spots. **d** Expression of *DNASE1L3* and complement components within the follicle, with spots coloured by expression and outlined as per classifications in (**b**). **e** Unsupervised clustering of spots using BayesSpace, with spots clustered based on gene expression and similarity to neighbouring spots. **f** Spots classified as lymphoid tissues (Cluster 7). **g** Sub-clustering of lymphoid tissue spots from (**f**) (GC = germinal centre). **h** Dot plot of gene expression within each subcluster, showing expression of cell lineage markers, *DNASE1L3* and complement components. **i** Correlation matrix of DNASE1L3, complement components and genes associated with apoptosis in lupus, showing correlation of expression within the lymphoid tissue spots in (**f**). Correlations were calculated as Spearman correlation with pairwise *p* values corrected for multiple inference using Holm's method, *p* values are detailed in Supplementary Fig. 3i. *$P < 0.05$; **$P < 0.01$; ***$P < 0.001$, ****$P < 0.0001$. Scalebars = 500 μm for (**a, b, d**), and 2 mm for (**e, f** and **g**). Source data are provided as a Source Data file.

C1q is less clear[24,53]. C1q can mediate bacterial interaction with host cells and is involved in the initiation of the classical pathway of complement activation that involves complement fixing antibodies[24]. Although IgG tends not to be secreted on mucosal surfaces in health, the other major complement fixing antibody class, IgM, is expressed[54]. Clones of IgM only memory cells in GALT and IgM secreting plasma cells in adjacent lamina propria are well described and an accepted feature of the intestinal immune system. Mucosal IgM specifically could be a mediator of bacterial killing in subepithelial regions following bacterial sampling by M cells.

Expression of *C1Q* has been identified as a marker of an embryonically derived subpopulation of macrophages in human lymphoid tissue that also express low levels of CD36 and *DNASE1L3*[43]. Our observation that *DNASE1L3* can be expressed by different subsets of macrophages and DC with contrasting phenotypes suggests that it may not contribute to lineage definition; rather this distribution supports inducibility[46]. It has been shown previously that DNASE1L3 can be induced by IL-4 and retinoic acid, and it is also already known that retinoic acid can be produced in the SED where it is involved in imprinting the expression of gut-homing receptors such as integrin α4β7[10,46]. It is thus possible that DNASE1L3 is also induced as part of a similar mucosal imprinting process.

DNASE1L3 and C1q (or more specifically their absence or inactivation) are notably associated with SLE and anti-DNA antibody formation[22,32,55,56]. The most cited antigenic driver of anti-DNA responses is DNA derived from apoptotic cells that is flipped to the external surface of dying cells[42]. In humans this is supported by the distribution of DNA on dying cells in vitro and there is also evidence that apoptotic debris generated by GC responses can be aberrantly distributed in the splenic GC of lupus patients[42,57]. However, here we show that DNASE1L3 is not expressed in the GC in human GALT, tonsil or lymph node but is located in the SED. DNASE1L3 is found both intra- and extracellularly, can digest bacterial DNA and even contribute to the disruption of biofilm[52]. Mouse models have contributed in many ways to the current view of the relationship between apoptosis and lupus pathogenesis, yet much data may not be translationally relevant because of the marked species difference in the distribution of DNASE1L3 that we identify here[58]. Thus, we propose that the role of apoptotic debris as drivers of anti-DNA responses in SLE may not be the full story, and that DNASE1L3 in the SED may have a role in preventing anti-DNA antibody production by DN2 cells after recognition of bacterial DNA.

TLR7 and TLR9 are both receptors for pathogen genetic material that are likely to be expressed in gut-associated lymphoid tissues, though expression levels were too low to detect their transcripts reliably in our data. They are also both associated with aberrant reponses to their ligands in lupus, though in different ways[59–61]. Whereas over expression of TLR7 results in more severe disease, TLR9 can be protective[61]. TLR9 recognition of CpG has shown to be involved in class switching to IgA[51]. This system that is driven by microbial ligands could be impacted by DNASE1L3 deficiency in lupus.

In conclusion, we describe a high prevalence of DN2 B cells that are known to be capable of T cell independent responses at the boundary between the microbiota and the host in GALT[20]. We also observe by spatial transcriptomics resolved to single cells that the most enriched transcript in FAE and SED encodes the lupus associated autoantigen DNASE1L3, which is expressed together with C1q. DNASE1L3 has a known role in degrading pathogen derived DNA[52]. In line with the uptake of bacterial antigens into the SED, we propose that DNASE1L3 and C1q have key functions managing bacterial antigens on the epithelial boundary. Their activity could prevent exposure of DN2 B cells to excessive microbial DNA which could potentially lead to anti-DNA responses.

## Methods
We confirm that this study complies with ethical regulations as detailed below.

### Human tissues
This study was approved by the UK Research Ethics Committees administered through the Integrated Research Application System. Cells and tissues were analysed with Research Ethics Committee (REC) approval number 11/LO/1274, London Camberwell St. Giles Research Ethics Committee.

Gut biopsies and paired blood samples used for Mass Cytometry were obtained with informed consent and REC approval from 5 healthy patients undergoing investigative colonoscopy who had no evidence of inflammation or other pathology. Mononuclear cells from gut biopsies and blood, were isolated as described previously and were stained on day of collection. These were 3 males aged 56, 56, 72 and 2 females aged 30 and 61.

For IMC, Visium and RNAscope analysis, formalin fixed, paraffin embedded (FFPE) archived tissues were obtained from anonymous patients undergoing appendicectomy, tonsillectomy or surgery for removal of colorectal cancer. FFPE blocks from normal colon ($n = 6$ donors), appendix ($n = 8$ donors), ileum ($n = 5$ donors), mesenteric lymph node ($n = 2$ donors), tonsil ($n = 2$ donors) were used. For donors that underwent colorectal cancer removal, multiple blocks from different sites were analysed as indicated in the text. In each case the tissues were histologically normal.

### Mouse tissue
Transverse sections of small intestinal Peyer's patches were cut from FFPE small intestine from five C57BL/6 mice obtained from Charles River, Sulzfeld, and archived FFPE embedded Peyer's patches from 2 CBA mice. No regulated procedures (including use of genetically altered animals) were carried out on the animals used for this study and so project-specific ethical approval was not required from the local Animal Welfare and Ethical Review Body (AWERB). All aspects of the housing, maintenance and culling of the mice were in accordance with the Animals (Scientific Procedures) Act 1986 and Amendment Regulations 2012.

### Mass cytometry
Mass cytometry antibodies were purchased pre-conjugated from Fluidigm or protein-free from suppliers indicated in Supplementary Table 1 and conjugated in-house using MaxPar labelling kit (Fluidigm) as per manufacturer's instructions. Viability staining was done as follows: $2 \times 10^6$ cells washed in cell-staining medium (C-SM, PBS with 0.5% BSA and 2 mM EDTA), incubated for 20 min in 1 mL of pre-diluted

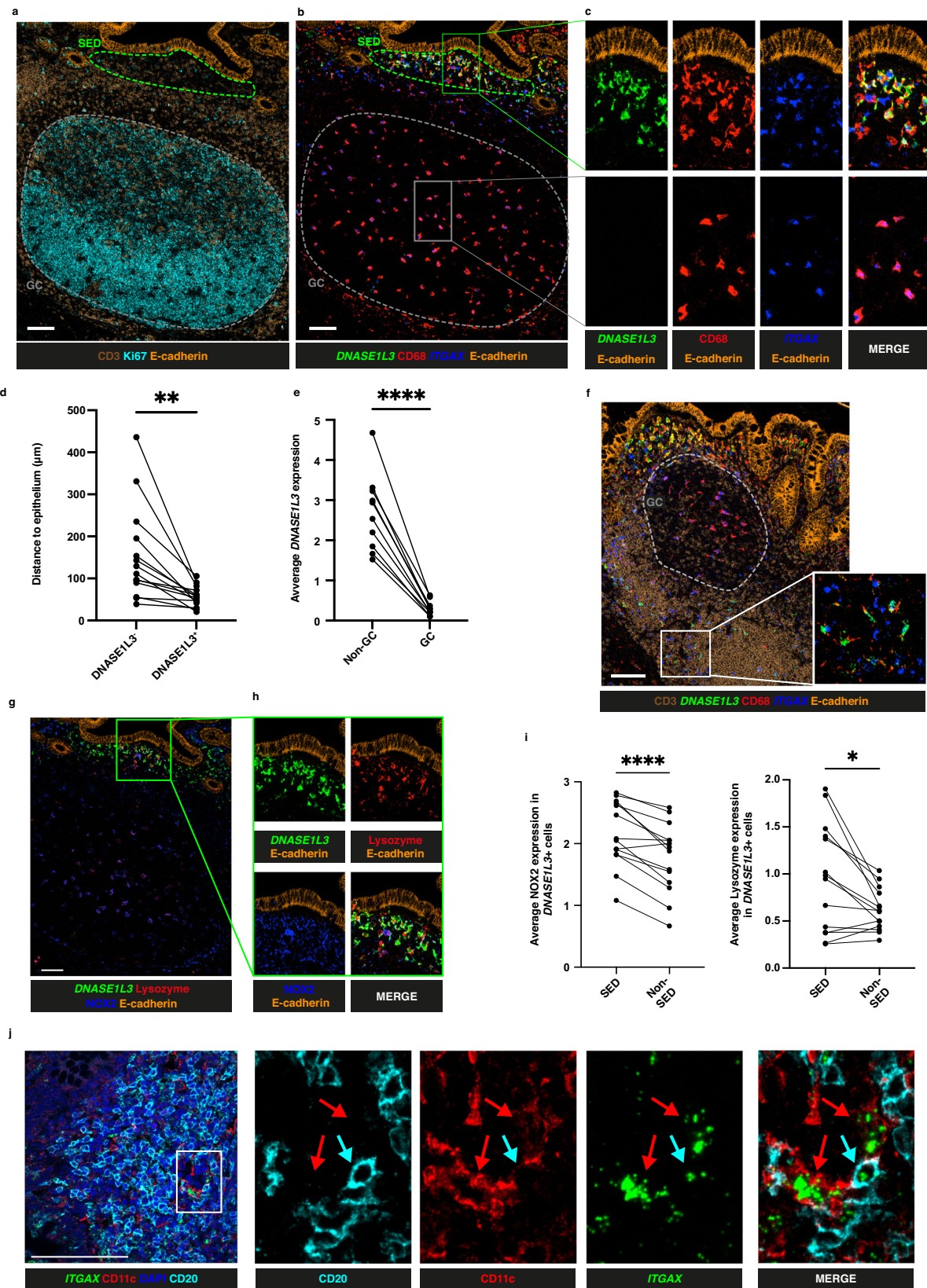

rhodium-intercalator (Fluidigm, 1:500 in PBS). After washing cells twice in C-SM, the cell pellet was resuspended in 10 μL Kiovig Fc block (Baxter, 5 mg/ mL) for 10 min at room temperature followed by the metal-conjugated titrated antibody cocktail in a total volume of 100 μL on ice for 30 min. After washing cells twice more in C-SM, cells were fixed overnight in 2% PFA in PBS at 4 °C. Cells were then centrifuged

and stained with 500 μl of 1 μM intercalating Ir (Fluidigm) in perm buffer (0.3% saponin [Fluidigm] in C-SM) and 100 μL intercalatin Ir (Fluidigm) for 30 min at room temperature. Finally, cells were washed twice with PBS and twice with Milli-Q water before resuspended at $0.5 \times 10^6$ cells/mL in Milli-Q water with EQ beads (DVS Sciences). Data was then immediately acquired using a Fluidigm Helios mass

**Fig. 4 | DNASE1L3 expression in the GALT subepithelial region. a** Imaging Mass Cytometry (IMC)-derived images illustrating spatial structure of an appendix follicle, including subepithelial dome (SED) (green dashed line) and germinal centre (GC) (grey dashed line), shown through expression of CD3 (T cells, brown), Ki67 (proliferation, cyan) and E-cadherin (epithelium, orange). **b** Same field as in (**a**) showing staining of CD68 (red) and E-cadherin (orange), as well as gene expression of *DNASE1L3* (green) and *ITGAX* (encoding for CD11c, blue) through RNAscope. **c** Enlarged profiles of SED and GC regions from (**b**) showing individual and merged marker expression. **d** Quantification of distance to epithelium for cells segmented on CD68 and *ITGAX* classified as either DNASE1L3⁺ or DNASE1L3⁻. $P = 0.0024$, $n = 14$ ROIs. **e** Quantification of *DNASE1L3* expression on cells segmented on CD68 and *ITGAX* classified as either GC or non-GC $P = 6.25899 \times 10^{-6}$, $n = 10$ ROIs. **f** IMC-derived images of ileum follicle illustrating CD3 (brown) and E-cadherin (orange) protein localisation alongside *DNASE1L3* (green) and *ITGAX* (blue) gene expression,

with GC highlighted (grey dashed line). **g** Visualisation of *DNASE1L3* (green), NOX2 (blue) and lysozyme (red) distribution within the same image as in (**a**). **h** Enlargement of SED region depicted in (**g**). **i** Quantification of the average expression of NOX2 and lysozyme in *DNASE1L3*⁺ cells and classified as residing in either the SED or in non-SED regions $P = 0.0107$ and $7.5789 \times 10^{-5}$ respectively, $n = 14$ ROIs. **j** Confocal microscopy image showing DAPI (blue), CD20 (cyan, pseudocolor), *ITGAX* (green) and CD11c (red) expression in an appendix follicle. *ITGAX*⁺CD20⁺ cells assessed using RNAscope are highlighted with cyan arrows. CD11c⁺*ITGAX*⁺CD20⁻ cells are highlighted with red arrows. Statistics between groups were assessed by two-tailed t test. *P < 0.05, **P < 0.01 and ****P < 0.0001. Scalebars = 100 μm. Findings shown in (**a**, **b**, **c**, **f**, **g**) are representative of 16 ROI. Data shown in (**j**) are representative of 6 ROI. Source data are provided as a Source Data file.

cytometer, and exported in.fcs format. The mild collagenase digestion of gut biopsies reduced the expression of TACI and CCR6 by mass cytometry; we excluded it from all further analysis.

## Analysis of mass cytometry data

FCS files loaded into Cytobank (https://mrc.cytobank.org/) underwent quality control and were filtered to live CD19+ cells only[13,15]. Equal numbers of events were extracted from each sample ($n = 9214$) and viSNE dimensionality reduction was run on the combined events using markers CD45RB, IgD, CD20, IgA, CD138, CD21, CD38, CD10, CD27, CD24, IgG and IgM. SPADE was used for clustering based on viSNE co-ordinates, and the resulting nodes were manually placed into B cell subset bubbles based on their expression of key markers (Fig. 1b). Marker expression, subset classification and viSNE coordinates were then exported and loaded into R for visualisation and quantification.

## Imaging mass cytometry

Sections were stained as described previously[62]. Briefly, slides were deparaffinised, rehydrated, and subjected to antigen retrieval using a pressure cooker. Tissues were then blocked and incubated with the mix of metal-conjugated antibodies specified in the panel overnight at 4 °C and subsequently incubated with the DNA intercalator Iridium Cell-ID Intercalator-Ir (Standard BioTools) before being air-dried. Slides were then inserted into the Hyperion Imaging System and photographed to aid region selection. Regions of about 1 mm² were selected and laser-ablated at 200-Hz frequency at 1 μm/pixel resolution.

## Analysis of imaging mass cytometry

Cell segmentation was performed following IMCSegmentation Pipeline[63]. Briefly, raw images for each ROI were extracted and a Random Forest Classifier was applied to each ROI based on manual labels of nucleus, cytoplasm and background on a set of training data[64]. Mean pixel intensities of each channel were calculated for each cell, and loaded into R, with analysis performed as per[65]. Intensities were arcsinh transformed for analysis, spillover correction of intensities was performed and quality control removed cells with size smaller than 11 pixels or with low counts of the DNA channels. Cells were phenotyped by a gating technique per image in the cytomapper R package[66]. Cells were gated as follows, with negative gates used to ensure mutually exclusive classifications: B cells (CD20 +), T cells (CD3 +), B-T neighbours (CD20 + CD3 +), Macrophage/DC (CD11c+ or CD11b+ or CD68 +), Endothelial (CD31 +) and Epithelial (E-cadherin +). Any cells in the B-T neighbours classification, or cells with more than 1 classification, were not given an assignment. A random forest classifier was then trained on the classified cells (split 70%/30% to training and testing data respectively) and applied to the unclassified cells in order to give each cell a phenotype[67]. Data were filtered to only cells classified as B cells. Batch correction of marker intensity values for B cells was performed using the Harmony package on a principal component analysis consisting of

main B cell markers[68]. Due to the presence of B cells within the epithelium, E-cadherin was included within the markers. Cells were then clustered, and a uniform manifold approximation and projection (UMAP) was performed. Clusters were manually classified based on their marker expression and visualised back on the images.

## Spatial transcriptomics

Two FFPE blocks (STA and STB) containing normal colon were used to perform Visium spatial transcriptomics according manufacturer's instructions as per demonstrated protocol CG000408 Revision A and CG000409 Revision A. Briefly, 6x6mm² sections from FFPE blocks were cut at 5 μm and placed on the Visium slide inside fiducial frames. Slides were then incubated for 2 h at 42 °C and H&E stained. After imaging using the S60 Nanozoomer scanner, the coverslip was removed, and slides were de-crosslinked. Human whole transcriptome probes (v1 chemistry), consisting of a pair of specific probes for ~18,000 genes, were added to each section for hybridising to their target genes. Hybridised and ligated probes were released from the tissue sections and captured on the spatially barcoded oligonucleotides on the capture areas. Sequencing libraries were generated from the probes and sequenced on the NextSeq 2000. Sequencing data was processed using Space Ranger pipelines (10x Genomics) for downstream analysis.

## Analysis of spatial transcriptomics

Transcriptomic counts of spots in each image were loaded into Seurat[28]. Using Loupe Browser 6.1.0 (10X Genomics), spots were manually annotated as belonging to the subepithelial dome (classification 'SED'), rest of lymphoid follicle (classification 'Follicle'), or outside of the follicle, and these manual cell annotations were attached to the Seurat objects. New Seurat objects using only spots in the SED and Follicle classifications were created and expression data for these were normalised using the SCTransform function[69]. Marker genes that were differentially expressed between the SED and Follicle classifications were determined using the FindMarkers function, with a Wilcoxon rank sum test and thresholds of 0.25 log-fold change and minimum expression in 10% of spots in either classification. Markers were visualised back on the images using custom R scripts.

The full set of spatial transcriptomic data was then analysed using the BayesSpace package in R, with raw counts log-normalised via the spatialPreprocess function, using 15 principal components and 2000 highly variable genes[70]. For spatial clustering, the number of clusters was determined by inspection of tuning plots created using the qTune function, with 7 clusters for sample STA and 8 clusters for sample STB. The spatialCluster function was then used to spatially cluster the data for each sample individually, with spots being assigned to clusters based on similarity of their transcriptomes along with a Markov Chain Monte Carlo algorithm using a spatial prior to encourage neighbouring spots to cluster together. For each sample, the cluster deemed to best represent lymphoid tissue was selected, and the Seurat objects were

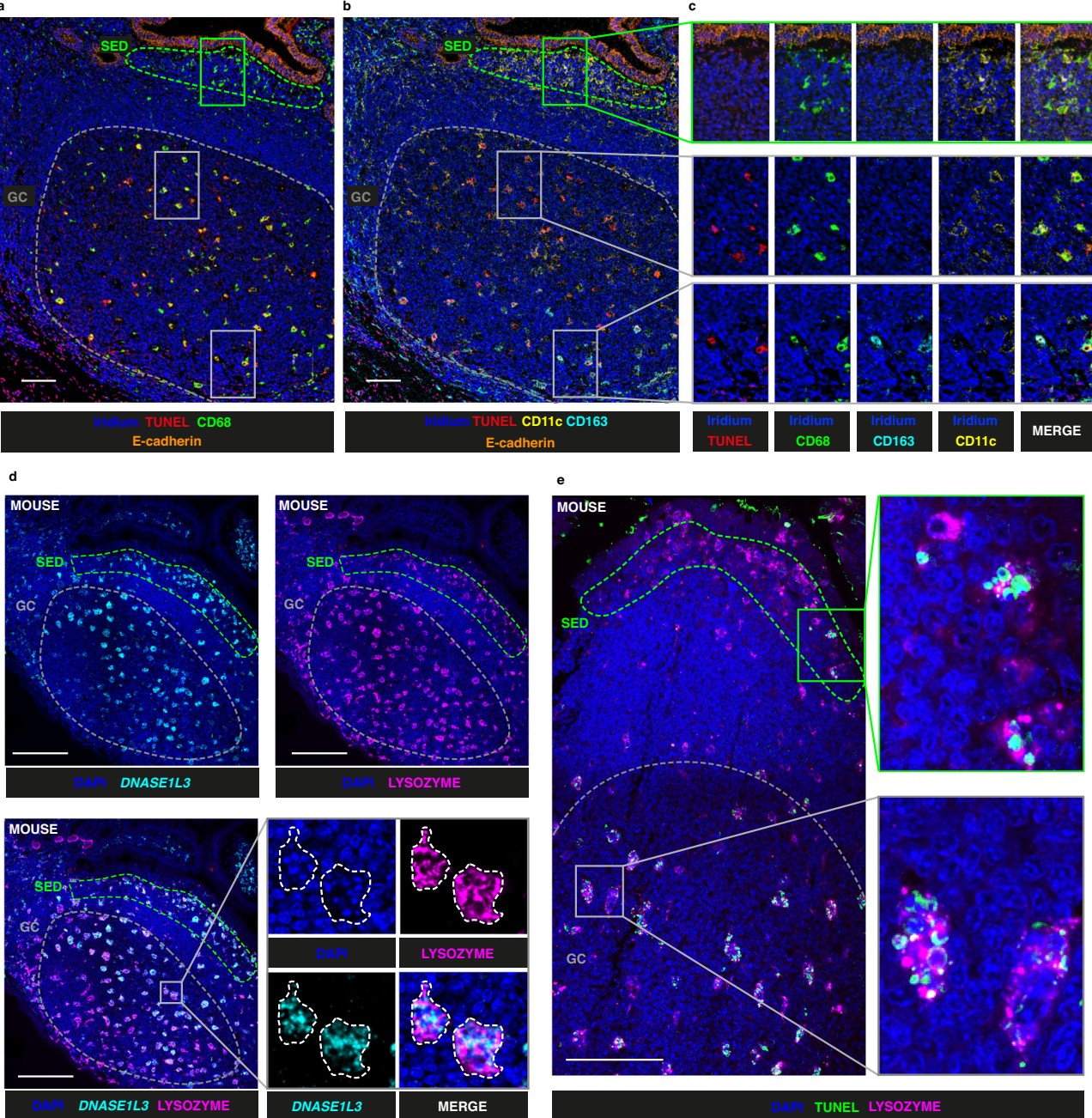

**Fig. 5 | Distribution of apoptotic debris and DNASE1L3 in human and mouse.**
**a** Visualisation of apoptotic debris (TUNEL, red), cell nuclei (Iridium, blue), CD68 (green) and E-cadherin (orange) obtained using an IMC-TUNEL combined protocol in a section serial to that shown in Fig. 4a. **b** Distribution of CD11c (yellow) and CD163 (cyan) on same image as in (**a**). Subepithelial dome (SED) and germinal centre (GC) are highlighted (green and grey dashed lines respectively).
**c** Magnification of regions in (**a**, **b**), showing individual and merged images for each marker. **d** RNAscope analysis showing the distribution of cell nuclei (DAPI, blue),

*DNASE1L3* transcripts (cyan, pseudocolor) and lysozyme protein (magenta, pseudocolor) in mouse GALT (Peyer's patch) assessed through confocal microscopy, including demarcation of the SED (green dashed line) and the GC (grey dashed line). **e** Visualisation of apoptotic debris with TUNEL assay (green) and lysozyme protein expression (magenta, pseudocolor) through confocal microscopy in the SED (green dashed line) and the GC (grey dashed line) of mouse GALT. Scale-bars = 100 μm. Findings for (**a**, **b**, **c**) are representative of 16 ROI. Findings shown in (**d**, **e**) are representative of $n = 5$ C57BL/6 mice and $n = 2$ CBA mice.

filtered to spots in these clusters. Data from both samples were merged, and a principal component analysis was run. Harmony was used to correct any batch effects between the samples[68]. Further sub-clustering of spots was done using this harmonised reduction, resulting in a set of sub-clusters present in both images. Marker genes for each sub-cluster were found using FindMarkers, and sub-clusters were relabelled based on their expression of cell lineage markers.

A list of SLE associated genes was collated, and correlations between the expression of each of these genes with the lymphoid

spots were calculated using the RcmdrMisc package[71] using Spearman correlation.

## RNAscope

RNAscope experiments were performed with Multiplex Fluorescent Reagent Kit V2 Assay (ACDBio #323110) following manufacturer's instructions on 5 μm thick FFPE sections. Briefly, slides were dewaxed blocked and subjected to antigen retrieval using a steamer for 22 min. Slides were then incubated at 40 °C for 30 min with the Protease Plus

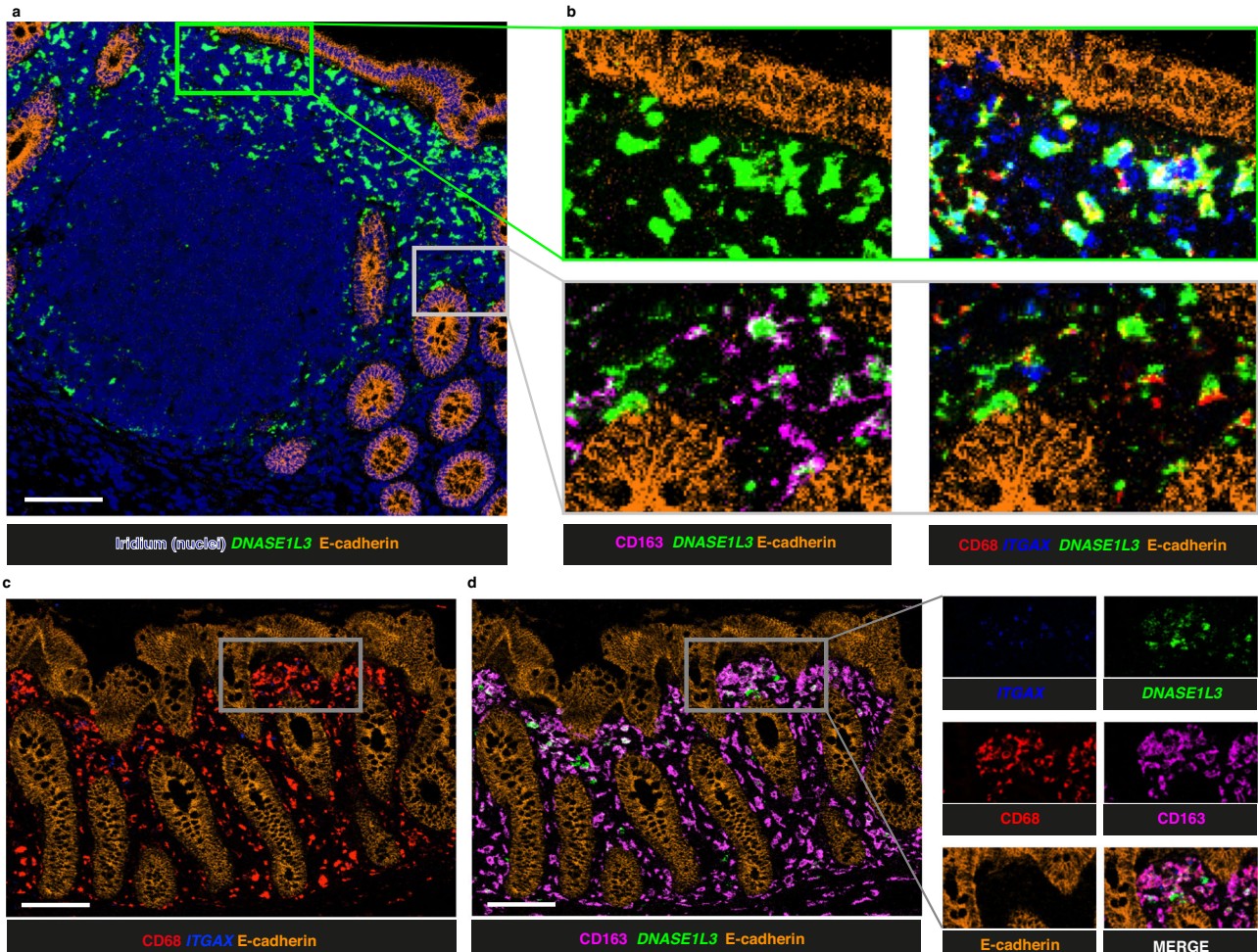

**Fig. 6 | Expression of ITGAX and DNASE1L3 in the colonic lamina propria.**
**a** Imaging Mass Cytometry (IMC) image of colonic folicle and adjacent lamina propria region showing the expression of *DNASE1L3* (green), E-cadherin (orange) and nuclear counterstaining Iridium (dark blue). **b** High power images of the follicle shown in (**a**), further illustrating the expression of CD163 (magenta), CD68 (red) and *ITGAX* (blue) in regions from the Subepithelial dome (SED) (green box) and

lamina propria (grey box). **c, d** IMC image of colonic lamina propria showing the distribution of E-cadherin (orange), CD68 (red), *ITGAX* (blue) alongside those of CD163 (magenta) and *DNASE1L3* (green). Scalebars =100 μm. Findings shown in figure (**a, b**) are representative of 16 ROI. Findings shown in **c** and **d** are representative of 4 ROI.

and for 2 h with the mix of probes. The appropriate HRP-tagged oligos were then incubated for 15 min followed by incubation with TSA-FITC (1:75, #NEL741001KT) or TSA Vivid Fluorophore 570 (1:1500, Bio-Techne #7526) or TSA Vivid Fluorophore 650 (1:1500, BioTechne #7527) diluted in TSA buffer. The enzymatic reaction was then blocked using HRP Blocker for 15 min. When more than one probe was assessed simultaneously, a different set of oligos was incubated followed by another TSA reagent. The following ACDbio probes were used in different combinations: human-C1qB (#565931), human-C1R (#508951-C2), human-C3 (#430701-C3), human-DNASE1L3 (#576121-C2 and 576121-C1), mouse-DNASE1L3 (#819171), Eubacteria 16S (#464461-C3) *B. fragilis* (#575441-C1) and *E. coli* (#433291-C1). Slides were finally mounted with Vectashield (VectorLab, H-1200) supplemented with DAPI diluted 1:250 (BioLegend, 422801) and coverslipped. Negative and positive control probes provided were also run on serial sections.

For the co-detection of RNA and protein, after the incubation with HRP Blocker, slides were washed, blocked and incubated in primary overnight at 4 °C. Primary antibodies used were: Vimentin 1:500 (Standard BioTools, #3143027D), Lysozyme 1:400 (Biolegend, #860001), CD11c 1:300 (Abcam, #ab216655) CD20 (Biolegend, #382802) 1:300. Slides were washed twice in PBS-T for 5 min and incubated for 1 h with anti-rabbit AlexaFluor555 (Invitrogen, #A32794)

or anti-rabbit AlexaFluor647 (Invitrogen, #A32795TR) or anti-mouse AlexaFluor647 (Invitrogen, #A32787) diluted 1:500. Slides were then washed and coverslipped as described above. A solution containing 5% BSA in PBS-T was used as blocking reagent and antibody diluent.

Slides were imaged using the Eclipse Ti-A Inverted confocal microscope using 20x, 40x, 60X or 100X objectives. For high-resolution imaging of bacterial DNA (Fig. 8h and Supplementary Movie 1, 2), Z-stacks were acquired with a 100X objective at 0.2 μm intervals to cover the full thickness of the section.

For the quantification, images from the different transcripts analysed were separated in single RGB channel and imported in CellProfiler[72]. Images were then made binary using the "Threshold" module and masked. The proportion of pixels masked was then divided by total pixels of the transcript of interest.

**IMC-RNAscope combined protocol**
Slides were processed as above for RNAscope protocol using DNA-SE1L3 (ACDbio, #576121-C1), ITGAX (ACDbio, #419151-C2) and Eubacteria 16S (#464461-C3) probes. C1 oligos were developed with TSA-Digoxygenin (Akoya Biosciences, NEL748001KT), C2 oligos with TSA-FITC, (Akoya Biosciences, #NEL741001KT) and C3 oligos with TSA-Biotin (Akoya Biosciences, NEL749A001KT) diluted 1:100 in TSA

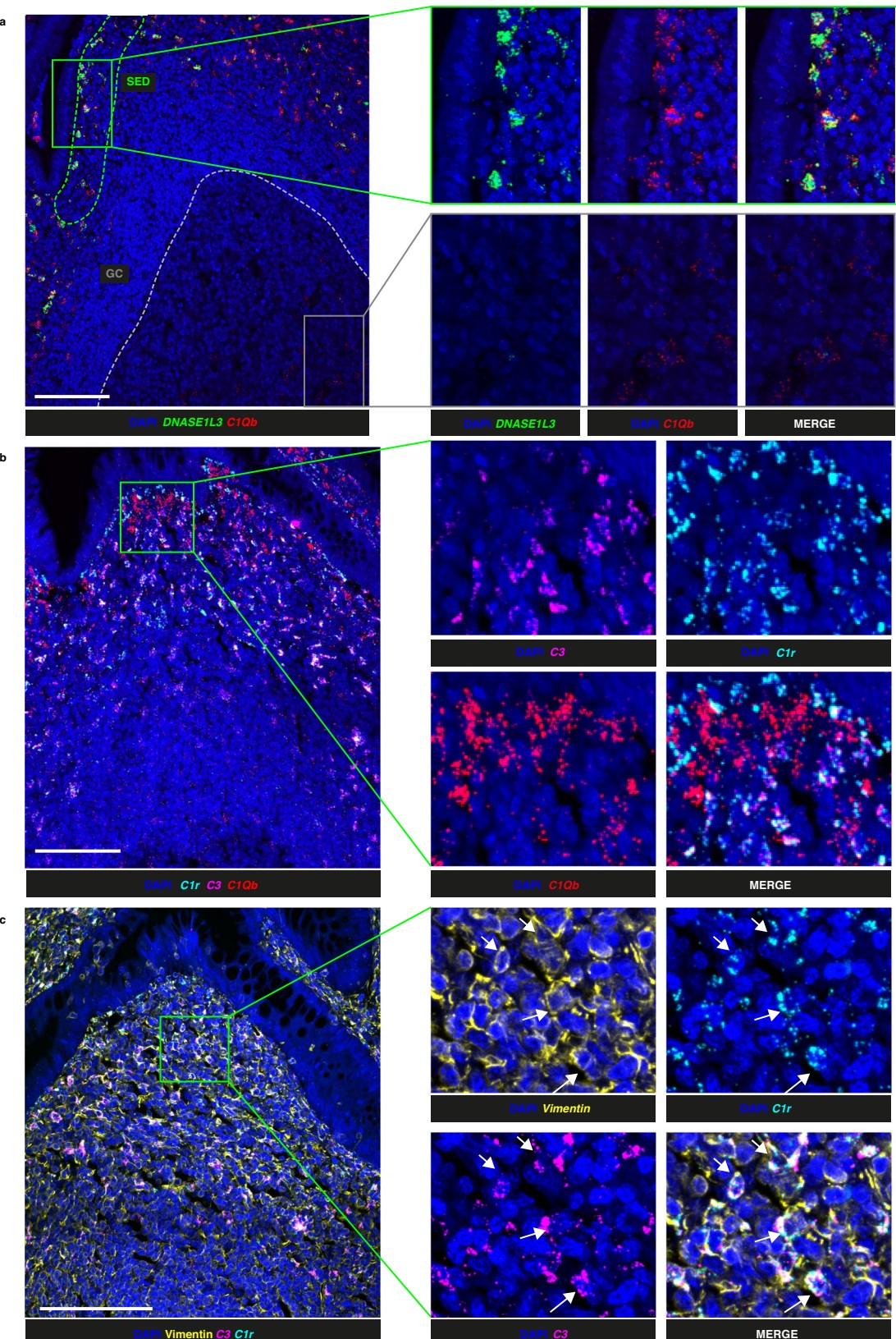

**Fig. 7 | Co-expression of C1Q with DNASE1L3 and C1R with C3. a** RNAscope coupled with confocal microscopy showing cell nuclei (DAPI, blue), *DNASE1L3* (green) and *C1Qb* (red) transcripts in the subepithelial dome (SED) (green dashed line) and germinal centre (GC) (grey dashed line) of an appendix follicle. **b** RNAscope coupled with confocal microscopy showing cell nuclei (DAPI, blue, pseudocolor), *C1r* (cyan, pseudocolor), *C3* (magenta, pseudocolor) and *C1Qb* (red,

pseudocolor) transcripts in an appendix follicle. **c** RNAscope coupled with confocal microscopy showing cell nuclei (DAPI, blue), Vimentin protein (yellow, pseudocolor), *C3* (magenta, pseudocolor) and *C1R* (cyan, pseudocolor) transcripts in a human appendix follicle. Scalebars =100 µm. Findings shown in (**a**, **b**, **c**) are representiative of 6, 9 and 4 ROI respectively.

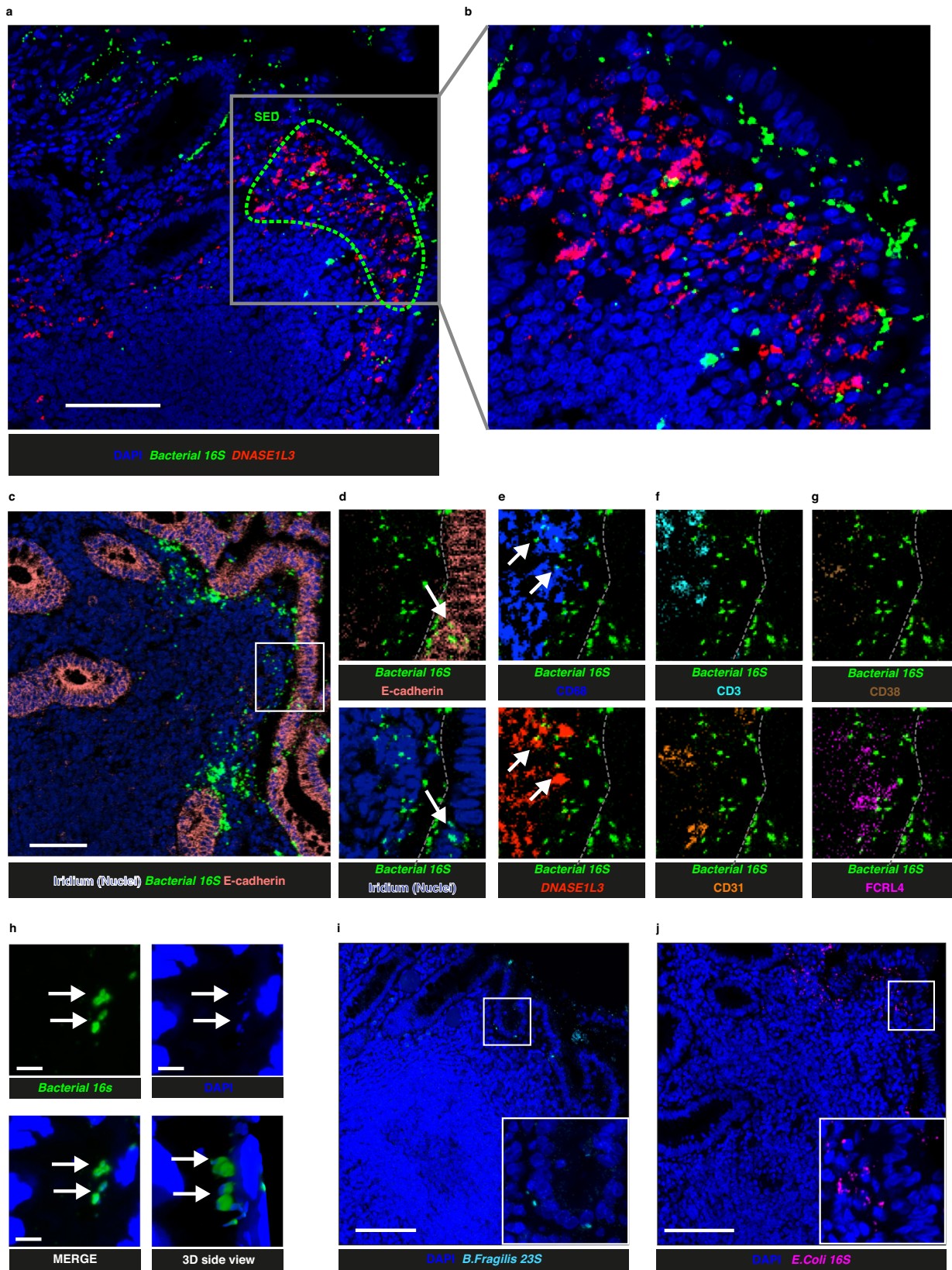

buffer. After the RNAscope protocol, slides were washed, blocked, and stained using Panel 2 and 3 (Supplementary Table 2) as described in the IMC paragraph above. Panels 2 and 3 incorporated anti-Digoxygenin (RD System, #611621), anti-FITC (Standard BioTools, #3174006B) and anti-Biotin (Biolegend, #409002) metal-tagged antibodies to detect the RNAscope staining output.

For quantification of co-expression of DNASE1L3 and Lysozyme or DNASE1L3 and NOX2 expressing cells, DNASE1L3+ cells were segmented using the FindPrimaryObjects module in CellProfiler using the DNASE1L3 channel as input[72]. The expression of all markers was calculated for each cell and arcsinh transformed, and cells were manually labelled as belonging to the SED using a custom R script.

**Fig. 8 | Localisation of internalised bacteria. a** Confocal microscopy images showing distribution of *DNASE1L3* (red, pseudocolor) and *Bacterial 16*S (green, pseudocolor) transcripts in a colonic follicle with subepithelial dome (SED) highlighted (green dashed line). Bacterial RNA was assessed using a probe against a region of the 16S transcript conserved across all Eubacteria by RNAscope. DAPI was used as nuclear counterstain. **b** Magnification of the image in (**a**). **c** RNAscope coupled with Imaging Mass Cytometry (IMC) analysis showing the distribution of *Bacterial* 16S, E-cadherin and nuclear counterstain (Iridium). **d–g** magnification of image shown in (**c**) alongside other markers for different cell lineages. **h** High-resolution confocal image showing *Bacterial* 16S (green, pseudocolor) distribution alongside DAPI (blue) staining in the SED of a representative gut-associated lymphoid tissue (GALT) follicle. Scalebar = 5 μm. **i, j** Confocal images of representative GALT follicles stained with RNAscope probes agains *B. fragilis* 23 s transcript (cyan, pseudocolor) and *E. coli* 16 S transcript (magenta, pseudocolor) respectively. DAPI was used as counterstain (blue). Scalebars = 100 μm except otherwise stated. Findings shown in (**a, b**) are representative of 6 ROI. Findings shown in (**c–g**) and (**h**) are representative 8 ROI. Findings shown in (**i, j**) are representative of 7 ROI.

### TUNEL assay and Lysozyme co-staining

TUNEL assay was performed using the In Situ Cell Death Detection Kit, Fluorescein (Roche, #11684795910) according to the manufacturers' instructions including an antigen retrieval in a pressure cooker with Basic Buffer (RD System, #CTS013). Slides were then blocked and incubated overnight at 4 °C with anti-Lysozyme antibody diluted 1:400 (Biolegend, #860001). Slides were washed and incubated with anti-rabbit Alexa Fluor 555 diluted 1:500 (Invitrogen, #A32794), coverslipped, and imaged as described above.

### IMC-TUNEL combined protocol

TUNEL assay was performed using the In Situ Cell Death Detection Kit, Fluorescein (Roche, #11684795910) as described above. Slides were then blocked and stained with Panel 1 (Supplementary Table 2) as described in the IMC paragraph above. An anti-FITC (Standard Bio-Tools, #3174006B) metal-tagged antibody was incorporated into the panel to detect the output of the TUNEL assay.

### Statistics

All statistical tests were done in Prism v9 unless stated otherwise. Two-tailed paired t tests were performed using parametric tests for comparisons of GALT and PBMC in mass cytometry data, as well as for paired comparisons of IMC data. For differential gene expression in spatial transcriptomic data, Wilcoxon tests with Bonferroni correction for multiple comparisons were used in the Seurat package in R version 4.2.2[28]. Statistically significant gene correlations in spatial transcriptomic data were identified using Spearman correlation in R along with pairwise *P* values corrected for multiple inference using Holm's method.

### Reporting summary

Further information on research design is available in the Nature Portfolio Reporting Summary linked to this article.

## Data availability

The spatial transcriptomic data generated in this study have been deposited in the GEO database under accession code GSE251693. The IMC image data generated in this study have been deposited on zenodo under record 10853309. Source data are provided with this paper.

## Code availability

Code used to perform analyses is available at github.com/jspencer-lab/GALT_analysis_2023 and have been deposited on zenodo under record 10849019.

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

## Acknowledgements

We are grateful to Emma Nye, Mary Green and Lucy Meader for the technical support in setting up the RNAscope IMC combined protocol, to Heli Vaikkinen and Kathy Fung for support with the spatial transcriptomics and to Jacqueline Siu for analysis of the mass cytometry data. This research was funded by the Wellcome Trust 220872/Z/20/Z (J.S.). This research was also supported by Research and Development, Guy's and St Thomas' NHS Foundation Trust. The views expressed are those of the author(s) and not necessarily those of the NHS.

## Author contributions

Project design and development: L.M., M.J.P., J.S., M.B., D.G., T.H.W. Carried out technical work: L.M., M.J.P., J.S., Y.Z., C.D., A.D., P.D., R.J.E., C.B., J.D.S., T.J.T. Method development and application: L.M., M.J.P., J.S., F.D.C., P.D., R.J.E., C.B. Data analysis and interpretation: M.J.P., L.M., J.S., M.B., D.G., S.J., D.D'.C. Wrote manuscript: L.M., M.J.P., J.S., M.B., D.G., T.J.T. Discussed and provided selected tissue: J.D.S., T.H.W., D.D.C., T.J.T., Raised funding: J.S.

## Competing interests

The authors declare no competing interests.
