## [Peer Review File · Nature Communications]

Double-negative B cells and DNASE1L3 colocalise with microbiota in gut associated lymphoid tissueREVIEWER COMMENTS

Reviewer #1 (Remarks to the Author):

This is an ambitious and well-conducted work Jo Spencer and co-workers employed multiple methodologies for microbiota data generation to investigate B cell signatures in GALT and their function. The work is impressive and conveys some important messages. There are also some caveats to reflect upon and consider in revisions.

Major comments

A major issue is that the connections to lupus nephritis (for example) are hypothesized on a very speculative basis, which make the conclusions far-fetched. I would advise to remove this link. Is it really needed for conveying your message?

Some other specific issues:

Title: It indicates that the study looks at the interaction of the microbiota with intestinal immunity and autoimmunity in the context of lupus nephritis. However, the method to look at bacterial signals is based on targeting Eubacteria 16S, which is wide and does not provide evidence of the function of the bacteria observed in a specific site.

Within this context, an experimental study testing the active migration of specific members of the microbiota to regions near GALT would be highly recommended.

The authors state that they “observe translocation of bacteria into GALT”. However, what the methodology allows them to determine is the presence of bacterial RNA near these locations. Other methods such as FISH would provide stronger evidence of the presence of whole bacterial cells and even give the possibility to use specific probes for specific microbes.

Minor comments

Introduction, line 62: APRIL also binds to BCMA.

Introduction, line 87: APRIL first explained here, but the abbreviation was also used earlier (line 62).

Reviewer #2 (Remarks to the Author):

The authors have presented a thorough interrogation of GALT with multiple complementary and justified cutting-edge technologies. I cannot comment on the significance of the author's findings in the context of the gut literature. However, there are no concerning analytical flaws in their presentation, with the analysis performed sensibly following well established pipelines, packages and workflows.

My only major comment is that I could not find any mention of how the authors intend to share their code or data. My apologies if I missed this.

Some minor concerns and commentary are listed below in no order:

On line 130, "Thus DN2/3 cells were enriched in GALT where their cell surface marker expression suggested epithelial proximity and T cell independence." It should probably be made clear that you aren't measuring these markers on the cell surface.

Arguably, some of the elements of the analytical strategies could have been made more consistent across the analysis ie. dimension reduction, cell type identification etc. This would have made the manuscript more cohesive and analytics simpler to explain and follow. However, these inconsistencies exist as the authors are applying established pipelines for each technology in isolation and shouldn't be criticised.

In the version I received there were multiple one or two sentence paragraphs. I am not sure if this is purposeful, but it did not flow well and should be revised.

Line 178 "heathmap"

Not ideal to use the same colour palletes across Figures 3e f and g.

Aesthetically and if motivated, some of the figures could benefit from some love in illustrator to make fonts and font sizes more consistent across panels within a figure.

Reviewer #3 (Remarks to the Author):

In this interesting report, the authors use multiplexed imaging technologies to identify a subpopulation of B cells that expresses DNASE1L3, a lupus autoantigen. They present data which shows that DNASE1L3 in the SED of the GALT is colocalized with bacteria and the complement component C1q (another lupus autoantigen). They suggest that their image analysis demonstrates that these lupus autoantigens are microbiota associated and typical of homeostatic intestinal immunity (and that this process goes awry in lupus). Although it is difficult to determine how many replicates are included in the data shown and what exactly they represent it is clear that the number of individuals or sections examined is very low for the broad conclusions made by the authors. The cutting-edge analyses performed are detailed and painstaking, but their significance and novelty depend on detecting the associations described in multiple sections from different tissues from multiple individuals. The conclusions drawn overinterpret the data presented from static images in a very limited sample set.

For Fig.1 the Methods (but not the legend) states that there were five donors that contributed to the mass spectrometry data. Five donors are clearly indicated in Fig. 1f. The legend says "donors" with respect to the pie chart in Fig. 1e. How many donors contributed to Fig. 1 a-d? Is it just one or a pool of all 5? If just one, which one - the individuals shown in Fig. 1f are quite different from each other.

As far as this reviewer can tell Fig. 2a-f represents a highly detailed analysis of one colonic section and 2g is a "representative" section of an appendix follicle, each from one person (and different from the five donors in Fig.1). How representative is this data? How many times were these findings reproduced, in how many individuals, with how many tissue sections? The text says that the authors analyzed paraffin sections of human GALT from

ileum, colon and appendix. One section of colon and appendix are shown in Fig. 2 (ileum is in Fig.S1f). The paragraph in the text concludes "Thus DN cells, in particular DN2, are found on the mucosal front line in the SED and FAE of GALT." Is this sweeping conclusion based on the three tissue sections shown? If not where is the rest of the data summarized?

Fig. 3 shows one section of colon although the text indicates that "n=5 GALT sites in 2 colon samples" were examined. From Fig. S2 the two colon sections appear to be STA and STB. Fig. 3 seems to be STA. The 5 GALT sites must be areas within these STA and STB sections. Again, how representative is the data in these two colon sections?

Fig.4 seems to show a detailed analysis of one section of appendix. One section each of appendix, ileum and colon are shown in Fig. S3. Are these sections taken from the same individual? The text says that regions of interest (ROI) from colon (n=7), ileum (n=4) and appendix (n=3) were examined. It seems that these ROIs are within these three sections.

Fig. 5a-c applies TUNEL analysis to a section serial to Fig. 2a. 5 d,e examines TUNEL staining in what is either one or two sections from a murine Peyer's patch. It is not clear what n=4 in the text means. The same concern applies to Fig. 6. Is examination of so few sections (possibly from one mouse) sufficient to conclude that there is a difference in the distribution of DNASE1L3 in mice and humans?

In Fig. 7 the authors return to an examination of appendix follicles to demonstrate colocalization of DNASE1L3 with complement components.

Finally in Fig. 8 the authors examine bacterial "uptake" and association with DNASE1L3. As expected, most of the bacteria detected are in the extracellular space. Again n=2 each for ileum, appendix and colon. Is this two ROIs or two sections?

The authors don't consider the role of the mucus layer in preventing access of bacteria to the SED. Have they tried to preserve the mucus? Can they stain it?

The authors state on p. 11 "This confirms that bacteria can cross human FAE and can be taken up by DNASE1L3+ DC in the SED or remain extracellular, though likely temporarily."

The static images presented don't demonstrate bacterial uptake, only some instances of colocalization. Again, are the very few samples examined sufficient to draw such broad conclusions?

Extending this analysis to lupus (or autoimmunity) is entirely speculative since no data from tissues of patients with autoimmune disease have been examined.

Minor point -

In the figures themselves the different sections are indicated by lower case letters (a,b,c...). In the figure legends the same sections are randomly indicated by both upper- and lower-case letters (a, B, C...). Please change all to lower case.

Responses to reviewers' comments

We are grateful for the time and thought the reviewers have given to assessment of our manuscript. We have made extensive changes including new figure parts, new supplementary figure, new supplementary movies, additional references and additions and clarifications to the text. In addition to tracked changes, we have highlighted modified text in yellow in the revised manuscript to help identification of them in the merged file. Where details of the changes are not transparent in the marked document, these are explained in further details below.

Reviewer #1 (Remarks to the Author):

This is an ambitious and well-conducted work Jo Spencer and co-workers employed multiple methodologies for microbiota data generation to investigate B cell signatures in GALT and their function. The work is impressive and conveys some important messages. There are also some caveats to reflect upon and consider in revisions.

We are pleased that the reviewer was impressed by what we submitted, and we are grateful for the opportunity to respond to their comments.

Major comments

A major issue is that the connections to lupus nephritis (for example) are hypothesized on a very speculative basis, which make the conclusions far-fetched. I would advise to remove this link. Is it really needed for conveying your message?

We accept this criticism and have toned down the speculative references to lupus nephritis, including removal of a paragraph from the discussion that was focused on autoimmunity and similarities to coeliac disease. However, we think some links are justified. For example, at the time of writing, a search for DN2 B cells on pubmed returns 43 entries. Of these 19 remain if lupus is also in the search criteria. Likewise, a search for DNASE1L3 returns 159 hits. Of these 54 remain if lupus is also in the search criteria, and these double hits for lupus and DNASE1L3 are focused in the most recent and high impact papers. That said, we accept the point the reviewer makes and have adjusted the text accordingly by removing speculative references to lupus pathogenesis. Text is removed from:

Abstract: line 45, 'lupus nephritis' replaced with 'autoimmunity'

Introduction: line 95, 'in lupus and Sjogren's disease' replaced by 'autoimmunity'. On line 101 'We suggest that these normal interacting features of GALT could be functionally caricatured or targeted aberrantly in lupus nephritis', has been deleted.

Results: line 256, 'that are otherwise associated with lupus nephritis' has been deleted. Also on line 256, 'lupus autoantigen' has been deleted. On line 344, 'double stranded' has been removed. On line 345, 'potentially' has been replaced by 'possibly'.

Discussion: deletion of the following paragraph, that was previously from line 442 onwards. Although we think this is a very interesting comparison, it is also speculative.

‘Coeliac disease and lupus nephritis are both autoimmune diseases where autoantigens (gliadin and DNA respectively) are substrates for enzymes (tissue transglutaminase 2 ([TG2]) and DNASE1L3 respectively), and where both the substrate and the enzyme are targets for antibodies ⁶⁰. Here we show that like coeliac disease, the enzyme and substrate in lupus nephritis are also associated with the gut epithelial boundary in GALT. In coeliac disease TG2 deamidates gliadin peptides derived from dietary gluten ⁶⁰. The resulting modified peptide complements the binding groove of HLA-DQ8 or HLA-DQ2 haplotypes from where it can be presented to gliadin specific T cells ⁶¹. B cells specific for gliadin can acquire help from T cells activated in this way resulting in the production of anti-gliadin antibodies ⁶². TG2 can also be taken up in complexes with gliadin and can itself then become target of an autoimmune response so that antibodies are made to both enzyme and substrate ^{62 63}. It is not so clear how tolerance is broken in the relationship between DNA and DNASE1L3. It was recently suggested that the main pathway for exposing the gut immune system to both gluten and TG2 would be through uptake from the gut lumen ⁶⁴. Future comparisons between lupus and coeliac disease could be helpful for the identification of potential parallel mechanisms, and potential T cell involvement in the breakage of tolerance.’

Some other specific issues:

Title: It indicates that the study looks at the interaction of the microbiota with intestinal immunity and autoimmunity in the context of lupus nephritis. However, the method to look at bacterial signals is based on targeting Eubacteria 16S, which is wide and does not provide evidence of the function of the bacteria observed in a specific site.

Thank you for this very good point. In response to this comment, we have now tested additional probes and can see *E. coli* and *B. fragilis* beneath the FAE with the same distribution as the Eubacteria 16S probe, though not as abundant which is logical. This is now described in the manuscript and included in a revised Fig. 8 and associated text, methods and legends.

Within this context, an experimental study testing the active migration of specific members of the microbiota to regions near GALT would be highly recommended.

Active sampling of the microbiota into GALT via M cells is known to occur in animal models (Rios, Wood et al. 2016). We agree that this would be a very interesting study to do in humans since our data refers exclusively to human GALT samples. As our paper emphasises there are many inter species differences, and it is important to study human tissues where we can. However unfortunately, such studies are not currently plausible. As yet, even organoids cannot replicate GALT. We apologise that we cannot respond

experimentally to this comment but have included new references in the text to support that this occurs in animal models.

The authors state that they “observe translocation of bacteria into GALT”. However, what the methodology allows them to determine is the presence of bacterial RNA near these locations.

We apologise for this over-statement. We acknowledge that we cannot observe translocation in static images and have adjusted the text.

Other methods such as FISH would provide stronger evidence of the presence of whole bacterial cells and even give the possibility to use specific probes for specific microbes.

We explored the possibility that whole bacterial cells could be present in our samples using higher resolution confocal microscopy. We were able to see examples where the Eubacteria 16s RNAScope probe detected RNA in cells that also contained DNA, with size and shape consistent with bacterial origin. This is now included in a revised Fig. 8h and new Supplementary Movies 1 and 2 where the bacterial signals identified using the probe can be visualised in 3D.

Minor comments

Introduction, line 62: APRIL also binds to BCMA.

Apologies for this mistake, this has been corrected.

Introduction, line 87: APRIL first explained here, but the abbreviation was also used earlier (line 62).

The text has been clarified so that APRIL is not introduced twice. We apologise for this mistake.

Reviewer #2 (Remarks to the Author):

The authors have presented a thorough interrogation of GALT with multiple complementary and justified cutting-edge technologies. I cannot comment on the significance of the author's findings in the context of the gut literature. However, there are no concerning analytical flaws in their presentation, with the analysis performed sensibly following well established pipelines, packages and workflows.

We are grateful for these comments and pleased that the reviewer appreciated our use of technologies and our analysis.

My only major comment is that I could not find any mention of how the authors intend to share their code or data. My apologies if I missed this.

We are sorry for missing this important information from the submission. We have now included in the manuscript that the code is accessible through GitHub and the spatial transcriptomic data is submitted to Geo. Both of these will be made live once the manuscript is published. Other datasets will be available on request.

Some minor concerns and commentary are listed below in no order:

On line 130, “Thus DN2/3 cells were enriched in GALT where their cell surface marker expression suggested epithelial proximity and T cell independence.” It should probably be made clear that you aren’t measuring these markers on the cell surface.

Apologies if we were unclear with this. The CyTOF method we used measured markers on the cell surface. For example, FcRL4 is expressed on the cell surface of B cells and was detected by antibody using CyTOF mass cytometry. It is known that high FcRL4 expression in the samples is seen spatially when cells are close to epithelium. We have rewritten to make this clearer.

Arguably, some of the elements of the analytical strategies could have been made more consistent across the analysis ie. dimension reduction, cell type identification etc. This would have made the manuscript more cohesive and analytics simpler to explain and follow. However, these inconsistencies exist as the authors are applying established pipelines for each technology in isolation and shouldn’t be criticised.

As the reviewer rightly reasons, not all our datasets lend themselves to the same analysis strategies. In each case we used the ones that in our view suited the application best.

In the version I received there were multiple one or two sentence paragraphs. I am not sure if this is purposeful, but it did not flow well and should be revised.

We joined paragraphs together where they were short and with information related to the content of adjacent paragraphs though the manuscript. We left the short summary paragraphs however at the end of the results sections.

Line 178 “heathmap”

Thank you for finding this typo that has now been corrected.

Not ideal to use the same colour palletes across Figures 3e f and g.

The colour scheme for Fig. 3 has been amended to use different colours for different panels.

Aesthetically and if motivated, some of the figures could benefit from some love in illustrator to make fonts and font sizes more consistent across panels within a figure.

Thank you for this suggestion. Fonts and font sizes have been made consistent where possible throughout the Figures.

Reviewer #3 (Remarks to the Author):

In this interesting report, the authors use multiplexed imaging technologies to identify a subpopulation of B cells that expresses DNASE1L3, a lupus autoantigen. They present data which shows that DNASE1L3 in the SED of the GALT is colocalized with bacteria and the complement component C1q (another lupus autoantigen). They suggest that their image analysis demonstrates that these lupus autoantigens are microbiota associated and typical of homeostatic intestinal immunity (and that this process goes awry in lupus).

We are grateful to this reviewer for their critique and suggestions. The reviewer made an error in the paragraph above that we assume is a typo. We did not study DNASE1L3 in B cells, rather we identified it in dendritic cells in the SED.

Although it is difficult to determine how many replicates are included in the data shown and what exactly they represent it is clear that the number of individuals or sections examined is very low for the broad conclusions made by the authors. The cutting-edge analyses performed are detailed and painstaking, but their significance and novelty depend on detecting the associations described in multiple sections from different tissues from multiple individuals. The conclusions drawn overinterpret the data presented from static images in a very limited sample set.

We are grateful to the reviewer for identifying that we had not provided sufficient information specifying how many donors and ROIs were used to derive data. This is complex because the human tissues used were required in specific orientations throughout the study and the numbers varied unavoidably. In general, we had decided to use exemplars in the figures that we now understand could give the impression that these were the only ones studied. Now, throughout the manuscript, we have stated how many donors, tissue sites, and how many ROIs were used for each piece of data acquisition.

For Fig.1 the Methods (but not the legend) states that there were five donors that contributed to the mass spectrometry data. Five donors are clearly indicated in Fig. 1f. The legend says "donors" with respect to the pie chart in Fig. 1e. How many donors contributed to Fig. 1 a-d? Is it just one or a pool of all 5? If just one, which one - the individuals shown in Fig. 1f are quite different from each other.

We thank the reviewer for noting that we had not stated how many individuals contributed data to each figure part. We have now clarified that all 5 sets of paired donor samples are included in a, b, d, e, f and g. The representative sample illustrated is c is now identified in f. An additional set of spade plots from another donor, also identified in Fig. 1f, is now included in a new Supplementary Fig. 1, and other supplementary figure numbers are adjusted accordingly.

As far as this reviewer can tell Fig. 2a-f represents a highly detailed analysis of one colonic section and 2g is a "representative" section of an appendix follicle, each from one person (and different from the five donors in Fig. 1). How representative is this data? How many times were these findings reproduced, in how many individuals, with how many tissue sections? The text says that the authors analyzed paraffin sections of human GALT from ileum, colon and appendix. One section of colon and appendix are shown in Fig. 2 (ileum is in Fig.S1f).

We apologise for not making the number of samples and ROI clear. Data analysed to generate the data in Fig. 2 were acquired from 5 donors in total (ileum= 3 donors, 5 ROI; appendix= 3 donors, 5 ROI; colon= 2 donors, 6 ROI). Typical examples were used for illustrative purposes in Fig. 2 and Supplementary Fig. 2. Once cells were segmented the cells from all tissues were pooled and underwent quality control before downstream analysis together. This is now explained in more detail in the text and modified Supplementary Fig. 2a,d,e.

In addition to the analysis already described we had also validated the presence of DN2 cells in epithelium observationally in additional samples of ileum and appendix from different individuals that had not been used for the computational analysis. This has also been included in the results section of the manuscript and Supplementary Fig. 2i. The 3 additional archived appendix samples have been added to the methods section.

The paragraph in the text concludes "Thus DN cells, in particular DN2, are found on the mucosal front line in the SED and FAE of GALT." Is this sweeping conclusion based on the three tissue sections shown? If not where is the rest of the data summarized?

The reviewer makes a very good point. We have now supported this statement with additional computational analysis and a new Fig. 2e. Other figure parts, text and the legend are adjusted accordingly. We hope that this not only demonstrates DN2 B cell proximity to epithelium, but it also illustrates more clearly that the outcomes are based on multiple datapoints and not on single sections.

Fig. 3 shows one section of colon although the text indicates that "n=5 GALT sites in 2 colon samples" were examined. From Fig. S2 the two colon sections appear to be STA and STB. Fig. 3 seems to be STA. The 5 GALT sites must be areas within these STA and STB sections. Again, how representative is the data in these two colon sections?

The reviewer is correct that 5 sites of GALT in two sections of colon were analysed here, and we agree that this data should not stand alone. We use the visium here as an undirected discovery tool and indeed it identified DNASE1L3 as a top hit in the SED, that was totally unanticipated. We think this illustrates the value of spatial transcriptomics. Critical to the point the reviewer makes, we then go on to investigate this signal in further donors, including additional sites of GALT, and at higher resolution and in the context of other cell types and bacteria.

Fig.4 seems to show a detailed analysis of one section of appendix. One section each of appendix, ileum and colon are shown in Fig. S3. Are these sections taken from the same individual?

We apologize for not making this clearer. The 4 ROIs shown in Fig. 4 and Supplementary Fig. 3 (now Supplementary Fig. 4) are from 3 different donors. The donor shown in Fig. 4a-c,g,h is the same as shown in Supplementary Fig. 3c while two different donors are analysed in Supplementary Fig. 3a and b.

The text says that regions of interest (ROI) from colon (n=7), ileum (n=4) and appendix (n=3) were examined. It seems that these ROIs are within these three sections.

Our apologies if this was unclear. These ROI were from 5 donors (ileum 3 donors, 4 ROI; appendix 3 donors, 5 ROI; colon 2 donors, 7 ROI). This has been clarified in the results section of the manuscript.

Fig. 5a-c applies TUNEL analysis to a section serial to Fig. 2a.

Fig. 5a is not from a serial section to that used for Fig. 2a. It is from a different ROI but from the same donor.

5 d,e examines TUNEL staining in what is either one or two sections from a murine Peyer's patch. It is not clear what n=4 in the text means.

The previous n=4 referred to 4 mice. We have now analysed 1 further C57BL/6 mice and 2 archived samples from CBA mice, all show the same profile. The extra samples have been included in the results and methods sections of the manuscript.

The same concern applies to Fig. 6. Is examination of so few sections (possibly from one mouse) sufficient to conclude that there is a difference in the distribution of DNASE1L3 in mice and humans?

The data in Fig. 6 is from humans, not mice. Fig. 6 described DNASE1L3 distribution in human intestinal lamina propria. Two observations were made here. The first related to the phenotype of DNASE1L3 in lamina propria that was consistent across 16 ROI from 6 donors. The second observation could only be made when the crypts were visible in

longitudinal section. This was possible in 4 ROI from 4 different donors. Because of the different numbers of samples that contributed to these observations, they have been illustrated separately in new Fig. 6. The text has also been updated to clarify the number of donors and ROI for the parts of Fig. 6.

In Fig. 7 the authors return to an examination of appendix follicles to demonstrate colocalization of DNASE1L3 with complement components.

The number of donors and ROI have now been added to the text.

Finally in Fig. 8 the authors examine bacterial "uptake" and association with DNASE1L3. As expected, most of the bacteria detected are in the extracellular space. Again n=2 each for ileum, appendix and colon. Is this two ROIs or two sections?

The number of donors and ROI have now been clarified within the text for Fig. 8.

The authors don't consider the role of the mucus layer in preventing access of bacteria to the SED. Have they tried to preserve the mucus? Can they stain it?

We agree with the reviewer that the mucus layer is important. We can't identify mucus in a meaningful way in the tissues available to us, but we can identify goblet cells using a periodic acid-Schiff (PAS) method that stains mucus pink in our FFPE samples (Figure 1, below). In the figure below we can see what appears to be goblet cell depletion over the FAE consistent with previous work (Merga, Campbell et al. 2014). However, this is a controversial field where not everyone agrees (Ermund, Gustafsson et al. 2013). On balance we think that this point would be too difficult for us to deal with in a meaningful way here. We have therefore not included this in the manuscript.

Figure 1: FFPE section of human appendix where we stained mucus pink using the Periodic acid-Schiff method. The FAE appears to be goblet cells depleted but it would be challenging and a distraction from the focus of the manuscript if we were to do this question justice.

The authors state on p. 11 "This confirms that bacteria can cross human FAE and can be taken up by DNASE1L3+ DC in the SED or remain extracellular, though likely temporarily." The static images presented don't demonstrate bacterial uptake, only some instances of co-localization.

We agree that we cannot infer active process from static images and have changed the text to more accurately represent what we see.

Again, are the very few samples examined sufficient to draw such broad conclusions?

We apologise that in the first submission we had been unclear when describing the number of samples studied. We had not made clear the extent of the work and validations that had been completed to arrive at our conclusions. It should also be noted that we ensured that all of the tissues we studied were well orientated. This has now been clarified throughout the manuscript in terms of donors, sites of GALT and ROI.

Extending this analysis to lupus (or autoimmunity) is entirely speculative since no data from tissues of patients with autoimmune disease have been examined.

We agree with the reviewer and have toned down the manuscript in this regard. However, we think some links are justified. For example, at the time of writing, a search for DN2 B cells on pubmed returns 43 entries. Of these 19 remain if lupus is also in the search criteria. Likewise, a search for DNASE1L3 returns 159 hits. Of these 54 remain if lupus is also in the search criteria. Importantly, the double hits for lupus and DNASE1L3 are focused in the most recent and high impact papers. That said, we accept the point the reviewer makes and have adjusted the text accordingly by removing speculative references to lupus pathogenesis. Text is replaced or removed from:

Abstract: line 45, 'lupus nephritis' replaced with 'autoimmunity'

Introduction: line 95, 'in lupus and Sjogren's disease' replaced by 'autoimmunity'. On line 101 'We suggest that these normal interacting features of GALT could be functionally caricatured or targeted aberrantly in lupus nephritis', has been deleted.

Results: line 256, 'that are otherwise associated with lupus nephritis' has been deleted. Also on line 256, 'lupus autoantigen' has been deleted. On line 344, 'double stranded' has been removed. On line 345, 'potentially' has been replaced by 'possibly'.

Discussion: deletion of the following paragraph, that was previously from line 442 onwards. Although we think this is a very interesting comparison, we agree that it is also highly speculative.

'Coeliac disease and lupus nephritis are both autoimmune diseases where autoantigens (gliadin and DNA respectively) are substrates for enzymes (tissue transglutaminase 2 ([TG2]) and DNASE1L3 respectively), and where both the substrate and the enzyme are targets for antibodies⁶⁰. Here we show that like coeliac disease, the enzyme and substrate

in lupus nephritis are also associated with the gut epithelial boundary in GALT. In coeliac disease TG2 deamidates gliadin peptides derived from dietary gluten⁶⁰. The resulting modified peptide complements the binding groove of HLA-DQ8 or HLA-DQ2 haplotypes from where it can be presented to gliadin specific T cells⁶¹. B cells specific for gliadin can acquire help from T cells activated in this way resulting in the production of anti-gliadin antibodies⁶². TG2 can also be taken up in complexes with gliadin and can itself then become target of an autoimmune response so that antibodies are made to both enzyme and substrate^{62,63}. It is not so clear how tolerance is broken in the relationship between DNA and DNASE1L3. It was recently suggested that the main pathway for exposing the gut immune system to both gluten and TG2 would be through uptake from the gut lumen⁶⁴. Future comparisons between lupus and coeliac disease could be helpful for the identification of potential parallel mechanisms, and potential T cell involvement in the breakage of tolerance.'

Minor point -

In the figures themselves the different sections are indicated by lower case letters (a,b,c...). In the figure legends the same sections are randomly indicated by both upper- and lower-case letters (a, B, C...). Please change all to lower case.

Thank you for pointing this out. This has been corrected.

References

- Ermund, A., J. K. Gustafsson, G. C. Hansson and A. V. Keita (2013). "Mucus properties and goblet cell quantification in mouse, rat and human ileal Peyer's patches." PLoS One **8**(12): e83688.
- Merga, Y., B. J. Campbell and J. M. Rhodes (2014). "Mucosal barrier, bacteria and inflammatory bowel disease: possibilities for therapy." Dig Dis **32**(4): 475-483.
- Rios, D., M. B. Wood, J. Li, B. Chassaing, A. T. Gewirtz and I. R. Williams (2016). "Antigen sampling by intestinal M cells is the principal pathway initiating mucosal IgA production to commensal enteric bacteria." Mucosal Immunol **9**(4): 907-916.

REVIEWERS' COMMENTS

Reviewer #1 (Remarks to the Author):

The authors responded to comments and criticism and performed additional work to address some of the reviewers' suggestions (albeit not all). The responses can be considered adequate and at this stage, I personally do not have any further comments or suggestions.

Reviewer #2 (Remarks to the Author):

The authors have addressed my comments appropriately. I have no further concerns.

Ellis Patrick

Reviewer #3 (Remarks to the Author):

The reviewers have been very responsive to the criticisms raised by the initial review. All concerns have been addressed.